# Hydrometeorological analysis and forecasting of a 3-day flash-flood-triggering desert rainstorm

Yair Rinat[1,*], Francesco Marra[1,2], Moshe Armon[1], Asher Metzger[1], Yoav Levi[3], Pavel Khain[3], Elyakom Vadislavsky[3], Marcelo Rosensaft[4], Efrat Morin[1]

(1) Institute of Earth Sciences, The Hebrew University of Jerusalem, Jerusalem, Israel
(2) National Research Council of Italy, Institute of Atmospheric Sciences and Climate, CNR-ISAC, Bologna, Italy
(3) Israel Meteorological Service, Beit Dagan, Israel
(4) Geological Survey of Israel, Jerusalem, Israel

*Correspondence to*: Yair Rinat (yair.rinat@mail.huji.ac.il)

**Abstract:** Flash floods are among the most devastating and lethal natural hazards. In 2018, three flash-flood episodes resulted in 46 casualties in the deserts of Israel and Jordan alone. This paper presents the hydrometeorological analysis and forecasting of a substantial storm (25–27 Apr 2018) that hit an arid desert basin (Zin, ~1400 $km^2$, southern Israel), claiming 12 human lives. This paper aims to: (a) spatially assess the severity of the storm, (b) quantify the time scale of the hydrological response, and (c) evaluate the available operational precipitation forecasting. Return periods of the storm's maximal rain intensities were derived locally, at 1-$km^2$ resolution, using weather radar data and a novel statistical methodology. A high-resolution grid-based hydrological model was used to study the intra-basin flash-flood magnitudes, which were consistent with direct information from witnesses. The model was further used to examine the hydrological response to different forecast scenarios. A small portion of the basin (1–20%) experienced extreme precipitation intensities (75- to 100-year return period), resulting in a local hydrological response of a high magnitude (10- to 50-year return period). Hillslope runoff, initiated minutes after the intense rainfall occurred, reached the streams, and resulted in peak discharge within tens of minutes. Available deterministic operational precipitation forecasts poorly predicted the hydrological response in the studied basins (tens to hundreds of $km^2$) mostly due to location inaccuracy. There was no gain from assimilating radar estimates in the numerical weather-prediction model. Therefore, we suggest using deterministic forecasts with caution as it might lead to fatal decision making. To cope with such errors a novel cost-effective methodology is applied by spatially shifting the forecasted precipitation fields. In this way, flash-flood occurrences were captured in most of the sub-basins, resulting in few false alarms.

## 1. Introduction

Flash floods are rapidly evolving events characterized by a sudden rise in stream-water level and discharge (Sene, 2013). These events often result in casualties and damage (Barredo, 2007; Borga et al., 2019; Doocy et al., 2013; Grodek et al., 2012; Petrucci et al., 2019; Vinet et al., 2019), and are ranked among the most devastating natural hazards worldwide (Barredo, 2007; Borga et al., 2014; Doocy et al., 2013; Gaume et al., 2009; Gruntfest and Handmer, 2001; Marchi et al., 2010; Sene, 2013).

Flash-flood conditions are frequently met in arid and semiarid regions (Nicholson, 2011; Pilgrim et al., 1988; Schick, 1988; Simmers, 2003; Tooth, 2000), as these areas are generally characterized by localized high rainfall intensities (Sharon, 1972), low precipitation interception and low infiltration rates due to sparse vegetation coverage (Danin, 1983), and exposed bedrock surfaces that are partially covered by shallow, clay-rich, undeveloped soils (Singer, 2007). Although arid and semiarid regions cover more than a third of the world's land area, knowledge of flash-flood-generating rainfall properties, hydrological response, and flood-forecasting skills in these areas is limited due to poor measurements, sparse documentation, and a relatively small number of studies (Armon et al., 2018; Nicholson, 2011; Simmers, 2003; Yang et al., 2017; Zoccatelli et al., 2019).

Rainstorm patterns in arid regions are characterized by localized structures of high rainfall intensities, termed convective rain cells (e.g., Karklinsky and Morin, 2006; Morin and Yakir, 2014; Nicholson, 2011; Sharon, 1972). During rainstorms, one or more convective rain cells can deliver relatively large rainfall amounts over small areas in a short time, directly contributing to runoff initiation and flash-flood occurrence (Archer et al., 2007; Borga et al., 2007; Chappell, 1986; Delrieu et al., 2005; Doswell et al., 1996; Gaume et al., 2016; Marchi et al., 2010; Yakir and Morin, 2011). Yakir and Morin (2011) found that flash floods in arid regions can occur as a result of a single rain cell, and that the flood's magnitude is sensitive to its starting location, direction, and velocity. Belachsen et al. (2017) found that, in arid regions, storms that generate flash floods are characterized by rain cells with larger area, lower advection velocity, and longer lifetime than storms that do not produce flash floods. Furthermore, rain-gauge networks are unable to adequately sample the spotty precipitation patterns (Faurès et al., 1995; Kampf et al., 2018; Michaud and Sorooshian, 1994; Wheater et al., 2007), limiting our knowledge of rainfall climatology and frequency (Marra et al., 2019b; Marra and Morin, 2015).

As a result of the complex rainfall patterns, runoff in arid and semiarid regions is often unexpected, localized, and characterized by high temporal variability (Morin et al., 2009b; Nicholson, 2011). Hillslope runoff does not always reach the stream network (Shmilovitz et al., 2020; Yair et al., 1980; Yair and Kossovsky, 2002; Yair and Raz-Yassif, 2004) and transmission losses can further enhance the intra-basin complexities and runoff localization (Greenbaum et al., 2002; Morin et al., 2009a; Walters, 1990). Even in the most extreme events, only part of the basin contributes runoff (Nicholson, 2011; Pilgrim et al., 1988; Yang et al., 2017).

The devastating effect of flash floods is attributed not only to the magnitude of the event, but also to their fast development and unexpected occurrence. When individuals or communities are not aware of the approaching danger, they are unable to

escape or protect themselves (e.g., Borga et al., 2019; Creutin et al., 2013). Thus, effective early warning of flash floods greatly depends on the time between the center of mass of the excess rainfall and the peak discharge (Dingman, 2015; USGS, 2012), termed lag time, and the time available between the issuing of the forecast and the peak discharge (Sene, 2013), termed lead time. The former is dictated by nature, while the latter also depends on the accuracy and effectiveness of the forecasting chain. Creutin et al. (2013) and Marchi et al. (2010) calculated lag times for Europe under various climate

regimes, and found that it increases with basin area and follows a general power-law behavior, where basins in areas smaller than 100 km$^2$ often had lag times of less than 1 h. Zoccatelli et al. (2019) found that the mean lag time for 14 arid basins (202–1232 km$^2$) in Israel is on the order of tens of minutes to several hours.

To increase flash-flood predictability and extend the lead time, accurate rainfall forecasting is required (Alfieri et al., 2012; Sene, 2013). Commonly used methods include weather-prediction models and nowcasting techniques. Global weather-

prediction models are routinely used by meteorological agencies worldwide, but their spatiotemporal scales are too coarse for flash-flood applications (Sene, 2013). In recent years, convection-permitting models with spatial resolution of ≤3 km have enabled explicit representation of the convective process, providing better representation of rainfall and better forecast skills on the flash-flood scale (Armon et al., 2020; Clark et al., 2016; Khain et al., 2019; Prein et al., 2015). However, the finer scale increases the sensitivity of these models to initial conditions, leading to spatial uncertainties in their output

(Bartsotas et al., 2016; Ben Bouallègue and Theis, 2014; Collier, 2007; Sivakumar, 2017). To cope with these limitations, radar rainfall estimates are routinely assimilated into the models (Clark et al., 2016; Stephan et al., 2008). Nevertheless, spatial and temporal uncertainties in individual forecasts are still observed, and multiple model runs should therefore be considered in a probabilistic ensemble framework (Ben Bouallègue and Theis, 2014; Dey et al., 2016).

In general, flash floods remain a poorly understood and documented process (Borga et al., 2019; Foody et al., 2004; Gaume

et al., 2009; Nicholson, 2011; Wheater et al., 2007) despite their devastation potential (Borga et al., 2019; Gaume et al., 2009; Inbar, 2019; Tarolli et al., 2012; Zekai, 2008) and increasing impact (Doocy et al., 2013; Wittenberg et al., 2007), especially in arid areas (Zoccatelli et al., 2019, 2020). The present work aims to increase our understanding and knowledge of desert flash floods and to test practical forecasting abilities by presenting a comprehensive study of the rainstorm of 25–27 Apr 2018 that hit the arid Zin basin in southern Israel (1400 km$^2$; Fig. 1), causing one of the most fatal desert flash floods

ever recorded in the region. The storm's highest impact occurred during the day of April 26[th], triggering a flash flood in the small (46 km$^2$) Zafit sub-basin (Fig. 1b,c). Rainstorm analysis was applied to all 3 days; however, its results and direct information from eyewitnesses led us to focus the hydrological and forecasting analyses mainly on the second day of the storm, April 26[th].

Specifically, we addressed the following questions: (a) What was the severity of the storm and flood, and how did it vary

spatially? (b) What was the time scale of the flash-flood response? (c) What was the operational predictability of the rainfall and the resulting flash floods, and can it be improved? To answer these questions, we combined datasets and tools, including

radar rainfall data, operational rainfall forecast, rainfall and flood-frequency analysis and their spatial variations, a grid-based hydrological model, and unique direct field observations during the event.

The paper is arranged as follows: the research area and data are presented in Sect. 2. Sect. 3 describes the rainstorm and
presents a spatial rainfall return period analysis. Sect. 4 and 5 focus on the hydrological response and operational forecast analysis of the flash-flood on 26 Apr 2018, respectively, Sect. 6 includes a short discussion, and Sect. 7 presents our conclusions.

## 2.    Study area and data

The Zin basin (~1400 km$^2$) drains the mountains of the Negev desert, southern Israel, to the Dead Sea (Fig. 1). Basin
orientation is from southwest to northeast, and elevation drops from 1000 m above sea level to 380 m below sea level. Slopes are low to moderate (0–10°) in 75% of the basin but can be as high as 60–80° locally (e.g., Fig. 1c). The main exposed lithology consists of limestone (54%), chalk and chert (27%), lithified sandstones (7%), non-lithified sand (5%), alluvium (4%), and marl (3%) (Sneh et al., 1998). The western part of the basin is covered by thin layers of lithosols that become scarce to the east, exposing the bedrock (Dan et al., 1975; Singer, 2007). Vegetation is extremely sparse, mainly
concentrated in stream channels, and its abundance decreases from west to east (Danin, 1983). We classified six hydrological domains in the basin according to the lithology, soil, and land use (Fig. 1b): rocky desert (80%), sand (6%), sandstone (5%), alluvium/colluvium (4%), quarry (3%), and built area (2%). Alluvial and sandy channel sections were identified as areas in which transmission losses occur (Greenbaum et al., 2002, 2006; Schick, 1988; Schwartz, 2001; Tooth, 2000; Wheater et al., 2007).

### 2.1.    Meteorological and hydrological setting

Mean annual rainfall ranges from 90 mm in the elevated western part of the basin to 60 mm in the lower eastern part. The rainy season spans October to May, with most of the rain (>60%) falling from December to February. Rainy days (≥1 mm) are rare and the annual average is 16 days on the western side and 8 days on the eastern side of the basin (IMS, 2020; averaged over 1980–2009). The mean annual potential evaporation is ~2600 mm; in the winter, it ranges from 2.6 to 4.6 mm
day$^{-1}$ in the western and eastern parts of the basin, respectively. In the autumn and spring, the mean potential evaporation spans 6.0 to 9.3 mm day$^{-1}$ with an increasing gradient from west to east (Goldreich, 2003).

Most flash floods (58%) occur during the winter months, from December to February, and 42% occur in the transition seasons (based on 107 flash-flood events recorded at the Zin Mapal station from 1954–2016; Fig. 1). Kahana et al. (2002) reported that most flash floods in the region can be attributed to well-defined synoptic systems. About a third of the events
are associated with Mediterranean Cyclones, occur mainly during the winter and include a wide range of magnitudes, and the rest occur during the transition seasons and are associated with flash floods of medium to extreme magnitudes. The maximal specific peak discharge measured at the Zin Mapal hydrometric station was 2.27 m$^3$ s$^{-1}$ km$^{-2}$ (October 1991, 234 km$^2$, Fig.

1b). The maximal specific peak discharge found in post-event surveys was 42 m$^3$ s$^{-1}$ km$^{-2}$ (October 2004, 0.5 km$^2$, with 12 m$^3$ s$^{-1}$ km$^{-2}$ in the Zafit sub-basin, 46 km$^2$; Fig. 1b).

## 2.2. Hydrometeorological data

The Zin basin is monitored by the Israel Meteorological Service (IMS) C-band Doppler radar, which provides scans at a temporal resolution of ~5 min and was used for the studied rainstorm. The area was also covered by the Shacham–Mekorot C-band radar, which has a 24-year-long record (operational up to 2015, see details in Marra and Morin (2015)) and was used for spatial rainfall-frequency analysis (Sect. 3.2). Both radars are located north of the basin (Fig. 1a). Two rain gauges with temporal resolution of 10-min and eight rain gauges that provide only daily data monitor the basin (Fig. 1a). Radar reflectivity data for the rainstorm were corrected for beam blockage and attenuation due to heavy rainfall. Rain intensity was calculated using a fixed Z–R relation well suited for convective precipitation in the area (Z = 316R$^{1.5}$) and converted to a 500 x 500 m$^2$ Cartesian grid (see appendix in Marra and Morin (2018)). Accumulated rain depths measured at 39 rain gauges in the basin's vicinity (Fig. 1a) were used to correct the residual bias for the analyzed storm using an adaptive multiquadratic surface-fitting algorithm (Amponsah et al., 2016; Martens et al., 2013). An upper cap of 150 mm h$^{-1}$ was applied on the adjusted estimates to reduce errors caused by the presence of hail (Marra and Morin, 2015). Class A evaporation pan measurements from Sde Boker (Fig. 1b) were used to evaluate daily potential evaporation rates.

The Zin basin includes four active hydrometric stations; however, only the Mamshit hydrometric station is situated at the area influenced by the storm's core (Fig. 1b, Table 1). To overcome the poor spatial representation of flood data, post-event surveys and analyses were conducted to estimate flood's peak discharge in nine ungauged locations (Fig. 1b, Table 1). High water marks were identified, channel cross sections were measured, and Manning roughness coefficients were estimated (Benson and Dalrymple, 1967; Gaume and Borga, 2008; Limerinos, 1970). Lastly, peak discharges were estimated using the HEC-RAS software (Brunner, 2016).

## 3. Rainstorm analysis

## 3.1. General storm description

The studied rainstorm occurred over 3 consecutive days from 25 to 27 Apr 2018, and covered most of the southern Israeli desert. It resulted from an upper-troposphere low, arriving to the eastern Mediterranean from the west. Moisture was available due to the slow passage of the cyclone over the Mediterranean Sea. This cyclone triggered highly developed convective clouds that roughly followed the upper-level low-pressure center (Dayan et al., 2020). Rainfall dynamics followed a similar pattern during all 3 days, initiating in the late morning (~10:00 all times are in UTC+3), and lasting intermittently until the evening (20:00–23:00) (Fig. 2a). Rainfall over the Zin basin was of a convective nature, characterized by 474 rain cells with mean rain-cell area of 237 km$^2$, mean areal rain intensity of 18 mm h$^{-1}$, mean maximal rain intensity of

57 mm h$^{-1}$, and mean rain-cell velocity of 16 m s$^{-1}$ (Fig. 2b; calculated over the Zin basin using a threshold of 5 mm h$^{-1}$ to determine rain-cell boundaries, following Belachsen et al. (2017)). The mean rain-cell area was similar to that estimated by Belachsen et al. (2017) for 29 flash-flood events in two basins draining into the Dead Sea (Fig. 1a), while mean and maximum rain intensity and cell velocity were, on average, higher in the current analyzed storm (Fig. 2b). This multiday convective rainstorm resulted in an extensive hydrological response, and flash floods were recorded at 27 of the 30 (90%) active hydrometric stations in the area of Fig. 2c.

### 3.2. Estimation of rainstorm return period

Estimating the probability of exceedance of the rainfall intensities observed during the storm is important to improve flash-flood warning systems, the design and operation of water resource projects, and risk and damage estimations for insurance policies (Brutsaert, 2005; Chow et al., 1988; Larsen et al., 2001). Traditionally, this information is derived from rain gauges, exploiting their relatively long and homogeneous records (Dey and Yan, 2016); however, especially in arid areas, rain gauges are generally sparsely distributed, resulting in an insufficient representation of the storm's spatiotemporal heterogeneity (Marra and Morin, 2018) on the one hand, and of the climatic gradients of the region on the other (Kidd et al., 2017; Marra and Morin, 2015).

Applying traditional approaches to the studied storm using one of the sub-daily rain gauge around the Zin basin (Sde Boker, Fig. 1a; Marra and Morin, 2015) results in return periods of less than a year for many durations (0.5, 1, 3, 24, and 72 h; not shown). However, this is misleading, as intense rainfall did not occur in this particular location during the storm. At the same time, a direct comparison of weather radar estimates to frequency analyses based on this station would be hampered by the strong climatic and topographic gradients characterizing the region (IMS, 2020; Marra et al., 2017): largely different frequency curves are to be expected within the Zin catchment due to gradients in both the intensity and frequency of occurrence of precipitation events (e.g., see Marra et al., 2019b). Therefore, different data sources should be explored.

Remotely sensed datasets, such as weather radar archives, may provide the required distributed information, and their use for precipitation-frequency analyses is becoming more and more quantitatively reliable due to the increasing length of the data records and improvements in the statistical techniques (Marra et al., 2019b). To obtain reasonable estimates of storm frequency for sub-daily durations and throughout the catchment, we took advantage of the 24-year-long Shaham–Mekorot weather radar archive (Marra and Morin, 2015), and of the novel metastatistical extreme value (MEV) framework (Marani and Ignaccolo, 2015). This latter method optimizes the use of short data records (Zorzetto et al., 2016) and is less sensitive to the measurement errors typical of weather radars than classical methods based on extreme value theory (Marra et al., 2018). Independent precipitation events were separated using the methodology detailed in Marra et al. (2018), and a single-event-type simplified MEV approach (Marra et al., 2019a) was used for the analyses, due to its robustness to the small number of rain events per year recorded in the area (Miniussi and Marani, 2020).

Maximum precipitation intensities observed during the storm for durations of 0.5, 1, 3, 6, 12, 24, and 72 h were derived for each radar pixel of the IMS radar, and spatial return period maps obtained by comparing these intensities to the above-described frequency curves based on the long radar archive are presented in Fig. 3a,b. Such return periods are to be interpreted as local, meaning that they represent the probability of exceeding the observed intensities in each pixel independently, and the gained output therefore offers a comprehensive spatial picture. Uncertainty related to the available data record was quantified via bootstrap with replacement (250 repetitions) among the years in the record (Overeem et al., 2008). We prudently estimate the return period of the observed amounts by relying on the 95[th] quantile of the bootstrap, meaning that the return period we communicate was exceeded with 95% probability. The spatial distribution of each return period in the Zin basin (Fig. 3d) and the timing of maximal intensity (Fig. 3e) were evaluated.

The spatial return period maps indicated two main areas characterized by extreme (75–100 years and 95% uncertainty range of 25–100 years) short-duration (0.5–1 h) intensities; one in the north-central part of the Zin basin and the other in the central part of the Zafit sub-basin (Fig. 3a,b). Conversely, only the central part of the Zafit sub-basin (Fig. 1b) experienced rainfall with long return periods for all durations (75–100 years for duration of 3–24 h and 10–25 years for duration of 72 h). One should consider that rain intensities over short durations might be biased due to interpretation of hail as intense rainfall, whereas, due to the temporal scale of convective cells, this problem should be negligible for longer durations (>3 h).

This application revealed high spatiotemporal heterogeneity of the rain intensities during the storm, clearly showing that using a single value for the entire region would lead to an incomplete, if not erroneous, interpretation. Even when observing the short durations (0.5–1 h), only 10–20% of the Zin basin experienced long return period intensities, while ~40% of the basin was characterized by return periods of 0–5 years (Fig. 3d). Rain-intensity timing maps (Fig. 3e) revealed that the high rain intensities in the northern part of the Zin basin and in the Zafit sub-basin occurred on different days: April 25[th] and 26[th] (12:00–24:00), respectively.

## 4.  Hydrological analysis

### 4.1.     The GB-HYDRA hydrological model application

GB-HYDRA is a high-resolution distributed hydrological model designed to study flash-flood dynamics in medium to small Mediterranean basins (Rinat et al., 2018). For the present study, additional components were included in the model to allow description of the arid environment (Fig. 4; Appendix A). The modified model was used at high spatiotemporal resolution (50 x 50 $m^2$; <60 s) to study the hydrological responses of 57 sub-basins (Table 2). Thus, propagation of various hydrological properties can be monitored, including identification of specific runoff-generating areas that directly contribute to stream discharge (referred to hereafter as runoff-contributing area [RCA]; see Rinat et al. (2018) for further details). Calibration results (see Appendix A for further details) pointed to adequate model performance and its use for this specific study ($R^2 = 0.94$; RMSD = 0.65 $m^3$ $s^{-1}$ $km^{-2}$; Bias = 0.35 $m^3$ $s^{-1}$ $km^{-2}$; Fig. 5, Table 1).

### 4.1.1. Using direct observations for spatial model validation and flash-flood initiation

A unique field observation from the second storm day (April 26[th]) provided a full description of runoff initiation in this arid environment, allowing us to obtain a spatial validation of the model. Two scientists from the Hebrew University of Jerusalem who were, coincidentally, at the Zafit sub-basin during the storm ("observation point" in Fig. 6b), fully documented and timed the processes of rainfall, hillslope runoff, and stream-discharge initiation. Their location and timing concurred with the radar-observed extreme rainfall intensities (see Sect. 3.2 for further details). Model simulations were validated against the observations, providing support for their validity. Agreement was found between field observations, weather radar estimates and model results for the timing of rainfall, RCA, and stream-runoff generation (Fig. 6, Table 3, Video 1).

### 4.2. Lag time

Basin lag time represents a simple, yet effective, way to estimate the basin hydrological response time. Excess rainfall was defined here as the difference between rain amounts and initial abstractions (following Marchi et al. (2010)), and rainfall separated by a hiatus greater than 1 h was not taken into account. A positive correlation between calculated lag time and basin area for each of the 3 days and for all sub-basins with peak discharge $>5$ m$^3$ s$^{-1}$ suggested that most of the calculated lag-time values were on the order of tens of minutes (Fig. 7). Finally, the calculated lag time for the Zafit flash flood on April 26[th] was 22 and 28 min for the simulation point and sub-basin outlet, respectively (Fig. 6). This short time emphasizes the difficulty in taking action after the rainfall starts, and the importance of an early warning.

### 4.3. Return periods of flash floods and their classification

Most sub-basins in the Zin catchment are not monitored, and therefore determining their flood return periods is not a trivial task (Haan, 2002). To overcome this, regional relations between specific peak discharges and basin areas were calculated to define categories of return periods. First, five categories of flash-flood return periods were determined: extreme (>50 years), large (10–50 years), moderate (2–10 years), low (<2 years), or no flow. Second, flood return period curves were built using generalized extreme value (GEV) analysis (applying the probability weighted moments method (Hosking et al., 1985)) and annual series of measured specific peak discharges for 18 hydrometric stations in the region (21–59 years, draining 60–3350 km$^2$, depending on the basin; data from four of these are shown in Fig. 8a; all stations are in arid environments). Third, the specific peak-discharge thresholds of the different categories (i.e., 2, 10 and 50 years) were computed for each station using the GEV curves (points in Fig. 8b). Fourth, a fit was applied to the calculated thresholds of each category (not shown). Fifth, final categories were determined by using percentiles of the eastern Mediterranean envelope curve (Tarolli et al., 2012) that match the fitted curves and the return periods detailed in stage 1 (schematic colored areas in Fig. 8b).

Using this analysis, the return period category of the April 26[th] (06:00-24:00) modeled peak discharge was defined for each sub-basin (Fig. 8c; Table 2). While moderate and large flash floods occurred in the eastern part of the Zin basin, no flash floods occurred in its western part.

## 5. Evaluation of flash-flood forecast

### 5.1. The COSMO numerical weather-prediction model

COSMO (COnsortium for Small-scale MOdeling) is a non-hydrostatic regional numerical weather prediction (NWP) model used by the IMS for operational forecasting (Baldauf et al., 2011; Doms et al., 2011). The model spatial resolution is of ~2.5 km, which enables explicitly resolving convective processes. Initial and boundary conditions are obtained from the integrated forecasting system (IFS) model run by the European Centre for Medium-Range Weather Forecasts (ECMWF). Ten-minute, real-time, rain gauge-corrected, 1 x 1 $km^2$ resolution, IMS radar rainfall data are assimilated into the COSMO model to improve its prediction (Stephan et al., 2008).

To assess and study the rainfall forecast's ability to predict such local events, different COSMO forecast runs were used as input to the GB-HYDRA model and the forecasted April 26[th] flash-flood category for each sub-basin was calculated. The IMS weather radar measurements, available in real time, were fed into the GB-HYDRA hydrological model and initial conditions were calculated. The resulting calculated flood categories were compared qualitatively and quantitatively to the flash-flood reference categories (Fig. 8c). For each run, the quantitative evaluation critical success index (CSI), false alarm ratio (FAR), probability of detection (POD), and probability of false detection (POFD) were calculated (Sene, 2013; Wilks, 2006). Flash-flood categories of moderate or above were used as the threshold.

### 5.2. Forecasted peak discharge

Two sets of COSMO rainfall-forecast runs produced using different methodologies of real-time radar measurement assimilation are presented in the following.

#### 5.2.1. Operational COSMO runs

Operational COSMO forecast runs were initiated every 12 h and assimilated with rain gauge-corrected radar data for the first 5 h, followed by a free run without any constraints. Rainfall data from five operational COSMO forecast runs were used (Apr 24, 09:30; Apr 24, 21:30; Apr 25, 09:30; Apr 25, 21:30; Apr 26, 09:30; timings indicate the approximate time of forecast availability; Fig. 9). Forecasted and measured accumulated rainfall fields for April 26[th] (06:00 to 24:00; Fig. 9) showed similar general patterns, even though location and timing might differ.

Forecasted flash-flood categories calculated for all sub-basins, based on applying the different rain forecasts to the GB-HYDRA model, are shown in Fig. 10. When qualitatively evaluated, all scenarios were found to differ from the reference

categories (Fig. 8c). CSI values were poor for all model runs (<0.25); moreover, in the first two scenarios, where the index was relatively high, the FAR index was high as well. It appeared that either the forecast does not result in flash-flood occurrence (Apr 25, 09:30; Apr 26, 09:30; Fig. 10), or flash floods are predicted in the wrong locations (Apr 24, 09:30; Apr 24, 21:30; Apr 25, 21:30; Fig. 10). In addition, the forecast skill did not improve with decreasing lead times.

### 5.2.2. **Ensemble of radar-assimilation scenarios**

The benefit of continuous radar assimilation into the COSMO weather-prediction model was tested for the two runs of April 25[th] at 21:30 and April 26[th] at 09:30. Thus, instead of applying radar assimilation for only the first 5 h, the COSMO model was re-run at hourly intervals, exploiting new radar data.

All runs were used as input to the GB-HYDRA model and the flood category in each sub-basin was calculated. No consistent improvement was found in either case with reduced lead time (Fig. 11). Forecasted flash-flood categories seemed to be random in both space and time. In fact, even the flood categories computed from COSMO rainfall for the hour in which the flash flood occurred at each sub-basin did not resemble the reference categories (Fig. 8c and right columns in Fig. 11). The forecast evaluation metrics showed poor results in all cases, with CSI values lower than 0.3, and high FAR values. For example, the flash floods at the Zafit sub-basins (Oretz, Mazar, and Zafit 1–3; Table 2, Fig. 11) were not predicted by most of the simulations.

### 5.3.    **COSMO spatial accuracy**

For the studied rainstorm, measured and forecasted rainfall shared a similar spotty pattern caused by convective rain cells of similar shape and size (Fig. 9). However, as already suggested, the forecasted rainfall might be erroneously placed in space and time (Ben Bouallègue and Theis, 2014; Collier, 2007). To account for this source of uncertainty, a simple and cost-effective forecast-shifting approach was applied: shifting the last two available COSMO runs closest to the April 26[th] flash-flood occurrence (i.e., Apr 25, 21:30, and Apr 26, 09:30) within a reasonable spatial error range of 20 km (Armon et al., 2020; Khain et al., 2019; Fig. 9) in 1-km intervals. For each shifted forecast, the 1681 resultant rainfall fields were used as input to the GB-HYDRA model. Then, for each sub-basin, the POD and POFD indexes were calculated using the modeled peak discharge and flood categories of April 26[th]. Shifting the COSMO run of Apr 25, 21:30 resulted in multiple flash-flood simulations of moderate or higher category in the eastern part of the Zin (Fig. 12a) and thus in a high range of POD values (0–80%) and low POFD (0–21%). The deterministic COSMO run of Apr 26 09:30 did not succeed in forecasting flash floods in any of the Zin sub-basins (Fig. 10). However, when shifted in space, it also resulted (Fig. 12b) in multiple flash-flood simulations of moderate or higher category in the eastern Zin basin (POD: 2–21%, and PODF: 0.2–8%), and presented a pattern similar to the reference categories (Fig. 8c).

## 6. Discussion

### 6.1. Analysis of precipitation severity in arid regions

Estimating the severity of flash flood producing storms is crucial to better manage the risks related to these natural hazards. The typical lack of data characterizing arid regions (Morin et al., 2020), together with the small scales of flash flood producing events (Borga et al., 2014), make the quantification of precipitation frequency curves at these areas extremely difficult. In this study we combined weather radar archives and novel statistical techniques, as suggested in Marra et al., (2019b). This enabled us to derive frequency curves for an ungauged area and produce a spatially distributed map of the event's severity which permitted to better understand its local impacts. In this way, a relatively short record of remotely sensed precipitation estimates could be fruitfully used to examine the local severity of a particularly hazardous storm. Thanks to the availability of high-resolution estimates from geostationary satellites, it is possible to extend these applications to arid regions of the globe for which no gauge or radar coverage is available, depending on the storms spatial scale.

### 6.2. Stream flow generation in arid regions

Runoff generation over hillslopes in arid regions is mainly controlled by infiltration excess (Simmers, 2003), which often occurs in response to intense rainfall (Nicholson, 2011). However, not all runoff turns into RCA and stream flow (e.g., Yair and Lavee, 1981), as rainfall duration is also a crucial factor (Vetter et al., 2014; Yair and Raz-Yassif, 2004). As a result, RCAs in arid regions are highly depended on rainfall properties (Shmilovitz et al., 2020; Vetter et al., 2014) and do not necessarily develop along streams (Bracken and Croke, 2007). Both field observations and model results indicates that in the studied rainstorm hillslope runoff is generated within minutes in response to intense rainfall. Two intense (>10 mm h$^{-1}$ in 5 min) rain showers separated by a 50-min dry period occurred at the observation point. The first lasted ~10 min and initiated local hillslope runoff but not RCA or stream runoff generation. The second lasted ~35 min and resulted in RCA generation and stream-runoff initiation within 10 min (Fig. 6, Table 3, Video 1).

Yakir and Morin (2011) and Morin and Yakir (2014) examined the sensitivity of simulated flash flood peak discharge to rain cell characteristics. They found that under most scenarios the maximal flood potential of a single rain cell is not fulfilled even when rainfall intensity is kept constant. Therefore, in most rainstorms, the flash flood return period is expected to be smaller than that of local rain intensities. In this work, the Zafit sub-basin experienced extreme rain intensities characterized by long return periods (75–100 years) on April 26[th] (Fig. 3b,e). However, the resultant flash flood was of large (10–50 years), but not of extreme magnitude. In fact, extreme rainfall covered only part of the sub-basin (Fig. 3b), and resulted in a maximum RCA value of 18% of the total sub-basin area (Fig. 6). This emphasizes the importance of RCA, together with rain intensity, for identifying flash-flood magnitude (Rinat et al., 2018).

## 6.3.    Flash flood forecasting in arid regions

Flash floods are a major challenge in arid regions due to their fast development (e.g., Marchi et al., 2010; Zoccatelli et al., 2019, 2020; Sect. 4.2) and potentially devastating impact (Gaume et al., 2009; Sene, 2013). Although, arid regions are not densely populated they are crossed by roads and attract tourists and hikers. On top of that, the ability to distribute forecasts and warning is often limited due to low cellular reception. The current low predictability of flash flood forecasts is dominated by the skill of the convective permitting rainfall-forecast models (Clark et al., 2016; Khain et al., 2019) and hydrological models (Collier, 2007; Moore et al., 2006; Wagener et al., 2007). In arid regions the possibility to improve these models using data assimilation and proper calibration is limited as monitoring instruments such as rain radars or rain and stream flow gauges are sparse (Kidd et al., 2017; Morin et al., 2020; Pilgrim et al., 1988) . Radar assimilation into the COSMO convective permitting model did not improve the forecasted hydrological response in the tested case study. This might be due to the rain cells' typically short lifetimes. According to Belachsen et al. (2017), the average lifetime of a convective flash-flood-producing rain cell around the Zin basin is ~40 min (median ~17 min). Thus, the hourly radar assimilations applied in this study are not likely to correct for convective rain cells that appear and decay within tens of minutes. Another reason may be that the assimilation algorithm does not effectively correct for cell location (Stephan et al., 2008).

Deterministic convection-permitting models fail to provide reliable forecast predictions at small spatial resolution (Roberts, 2008). Therefore, probabilistic approaches are applied to produce forecast ensembles (Ben Bouallègue and Theis, 2014; Clark et al., 2016). Forecast ensembles have proven to yield better results than deterministic ones (Alfieri et al., 2015; Liechti and Zappa, 2019), and are in operational use in the meteorological offices of several countries (e.g., DWD, 2020; Met Office, 2020; MeteoSwiss, 2020). The ensemble members are created by applying small perturbations to initial conditions or by varying the description of the physical process of the weather model (Hagelin et al., 2017). However, these methods require multiple model runs and intense computational resources (Clark et al., 2016) and therefore, less costly methods have been developed. These include smoothing, upscaling, or "neighborhood" methods (Ben Bouallègue and Theis, 2014; Schwartz and Sobash, 2017; Sobash et al., 2011; Theis et al., 2005). Vincendon et al. (2011) developed a simplified method to produce rainfall ensemble from single-value meteorological forecasts and showed that by using it as an input to a hydrological model the gained flood-forecasting ensemble performs better than the deterministic result. In this work we present a simple and low-computational-cost ensemble method using forecast-shifting that have the potential to improve flash-flood forecasting; however, to assess its overall benefit it should be furthered examined for a large variety of conditions.

Finally, as a complementary tool for flash floods prediction the testing of nowcasting techniques is recommended. Nowcasting methods apply various algorithms to propagate real-time rain radar observations, while maintaining the high spatiotemporal resolution of the original data (Sene, 2013). The extrapolated rain fields are used as input to a hydrological model thus allowing for short term (up to 6 h) hydrological predication (Alfieri et al., 2015; Berenguer et al., 2005; Sempere-

Torres et al., 2005). Additional methods assimilate the nowcasting products into convection-permitting models thus gaining longer rainfall forecast range (Sokol and Zacharov, 2012). Although, the accuracy of nowcasting is affected by several factors, for example: the short life time of convective rain cells, or the effect of orographic enhancement, it was found to significantly improve the flash flood forecasts in several case studies and to provide extended warning of 10-80 minutes (Berenguer et al., 2005). This relatively short contribution is not to be overlooked, especially when summed with the basin

lag time, found in this work to be of the order of tens of minutes.

## 7.  Conclusions

Knowledge of desert rainstorms and flash floods is limited, despite their devastating potential. To enrich this knowledge, we presented a comprehensive study of rainfall severity, hydrological impact, and forecasting potential for a fatal 3-day desert rainstorm. Special focus was placed on the storm's highest impact, which occurred on April 26[th], in the Zafit sub-basin.

The main conclusions of this study are:

- Rainfall-intensity return periods calculated at radar pixel resolution were presented. Using a novel method, we identified rain intensities of exceptionally long return periods (75–100 years, for most durations) on April 26[th] at the Zafit sub-basin, whereas for most of the Zin basin (>37–89%), the calculated return periods are 0–5 years.

- Rainfall and hydrological response in desert areas is local in nature. Despite the extreme (75-100 years) rainfall

intensities at the Zafit sub-basin, only a small part of it (20%) contributed runoff to stream discharge, resulting in large flash-flood estimation (return period of 10–50 years). In addition, around 35% of the total Zin basin did not experience any flash floods on April 26[th].

- Flash floods in desert areas develop quickly. Hillslope runoff is initiated within minutes and stream discharge within tens of minutes. Calculated lag times for the Zin sub-basins are on the order of tens of minutes as well.

• The use of current deterministic operational forecast models is insufficient for flash-flood forecasting in small to moderate desert basins. None of the five deterministic COSMO forecast runs that were available prior to April 26[th] managed to capture the flash-flood occurrence or magnitude. Radar assimilation did not improve the forecast results. However, simple spatial shifting of the deterministic forecasted rainfall led to improved probability of detecting the flash floods.

This single case study demonstrated the high potential for improving lifesaving flash floods forecasts. Comprehensive work on other events and other locations together with advanced nowcasting and ensemble prediction methods may benefit worldwide.

# 8. Appendix

## 8.1. GB-HYDRA application

GB-HYDRA is an event-based, distributed hydrological model developed to study medium to small Mediterranean basins. The formulation presented in Rinat et al. (2018) was slightly altered to fit the studied arid basins and now includes a description of the following hydrological processes: evaporation, infiltration and reinfiltration, hillslope and stream runoff, downward percolation, and transmission losses (Fig. 4). In addition, the model calculates the runoff-contributing area (RCA) at any given time, defined here as hillslope sections from which water flows and reaches the stream network, during a specific duration (defined in this work as 30 min following Morin et al. (2001)).

The ArcMap GIS program was used to prepare the spatial data, including topography, stream network, soil, and land use. A smoothing procedure was applied to correct and eliminate artificial jumps in the stream profiles, and reduce numerical instabilities (Peckham, 2009).

High-resolution aerial photography and 1:50,000 scale geological maps (Avni et al., 2016; Avni and Weiler, 2013; Hirsch, 1995; Roded, 1982, 1996; Yechieli et al., 1994; Zilberman and Avni, 2004) were used to identify alluvial stream sections, and areas of different runoff potential. Channel widths were measured in the field and estimated using the air photos at different locations along the Zin's main channel and tributaries. These widths were extrapolated by fitting a power-law function (Montgomery and Gran, 2001) between the measured width (dependent variable) and the drainage area (independent variable).

The initial soil water storage was set to zero, as no significant amount of rainfall had precipitated in the study area since the end of February 2018 (total rain depth <3.2 mm in 2 months). The daily potential evaporation rate measured at the Sde Boker station (Fig. 1b) was ~4 mm day$^{-1}$ during the storm. The Manning coefficient along the stream channels was estimated at 0.03, 0.04, and 0.05 for: no, coarse, and very coarse alluvial stream bedding, respectively (Shamir et al., 2013). Hillslope Manning roughness coefficients varied from 0.01 to 0.02 (Table A1) following Downer and Ogden (2002), Engman (1986), Sadeh et al. (2018), and Shmilovitz et al. (2020)). Stream sections characterized by alluvial bed were identified as areas prone to transmission losses (Fig. 1b) and assigned a constant infiltration rate of 10 mm h$^{-1}$ following Greenbaum et al. (2002), Lange (2005), Lange et al. (1999), and Morin et al. (2009).

Downward percolation of water from hillslope grid cells to underground storage was set to zero, except for grid cells identified as sands (Fig. 1b). These sections are characterized by the thick (up to 40 m) unconsolidated sandstones of the Hatzeva group (Calvo et al., 2001) and cover about 88 km$^2$ (~6%) of the Zin basin. A constant drainage rate of 40 mm h$^{-1}$ was used in these sections (Table A1) following Lange et al. (1999) and Lange and Leibundgut (2005). Infiltration rate was described by the Soil Conservation Service (SCS) conceptual method. Initial abstractions of 20% of the maximal soil storage were applied to each pixel (Chow et al., 1988; SCS, 1972).

In contrast to humid regions (Engman, 1986; Mishra and Singh, 2003), there are a limited number of studies utilizing the SCS method in arid and semiarid environments (Lange et al., 1999; Nouh, 1990; Shammout et al., 2018; Wheater et al., 2007; Zekai, 2008). Thus, arid and semiarid curve number (CN) values are still not firmly established. CN parameters of three land-cover types in the studied basin that are poorly represented in the scientific literature (sands, sandstones, and rocky desert; Table A1, Fig. 1b) were found by calibration. To save on computation time, calibration was performed only in

the eastern part of the Zin basin. This part of the basin includes the area of the storm's core and most of the available peak-discharge measurements. Initial CN values were determined from the literature (Moawad, 2013; Moawad et al., 2016; Wheater et al., 2007; Zekai, 2008). Root mean square difference (RMSD) of observed and computed specific peak discharges and their bias were used as objective functions, and final parameters were selected from the Pareto solution group (i.e., the group of parameter sets for which none provides better results than the other in terms of both RMSD and bias).

Calibration results pointed to adequate model performance and its use for this specific study ($R^2 = 0.94$; RMSD = 0.65 $m^3$ $s^{-1}$ $km^{-2}$; Bias = 0.35 $m^3$ $s^{-1}$ $km^{-2}$; Fig. 5, Table 1). Rinat et al. (2018) found that the sensitivity of the GB-HYDRA-modeled peak discharge and RCA to stream Manning roughness, hillslope Manning roughness, and CN values is low. In this work, sensitivity analysis was not applied due to considerably long run times. Further procedures of sensitivity analysis, calibration, and validation are needed if the model or its parameters are to be used in more general studies.

## 9.   Video supplement

Video 1: The video related to this article can be found at: https://doi.org/10.5446/47152

**Author contributions.** YR and EM conceptualised the work. The methodology and software were developed by YR and FM. Data curation and formal analysis were performed by YR, FM, AM, MA, YL, PK, and EV. Resources were provided by MR, YL, PK, and EV. Funding was acquired by EM who also supervised the work. YR wrote the original draft of this paper, which was reviewed and edited by all authors.

**Competing interests.** The authors declare that they have no conflict of interest.

**Acknowledgements**

The authors wish to thank the editor, Maria-Carmen Llasat, and the reviewers Lorenzo Marchi and one anonymous reviewer for their useful comments that helped improve the manuscript. We wish to thank Tamir Grodek, Alon Ronen, and the Soil Erosion Research Station for their help in post-event analysis peak discharge estimations. This study was funded by the Israel Science Foundation [grant no. 1069/18], a NSF–BSF grant [grant no. BSF 2016953], and a Google gift grant. This study is a contribution to the PALEX project "Paleohydrology and Extreme Floods from the Dead Sea ICDP Core" funded by the DFG [BR2208/13-1/-2], and is a contribution to the HyMeX program. We are also grateful for the detailed direct rainfall and flood observations made by Prof. Ari Matmon and Nathalie Neagu, and for sharing the information with us.

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

Table 1: Measured and modeled peak discharges for the Zin case study. Uncertainly limits refer to error estimations in the measured peak discharge

| # | Sub-basin name | Area (km$^2$) | Post event analysis (P) \ Hydrometric station (H) | Storm day | Measured peak discharge (m$^3$ s$^{-1}$ km$^{-2}$) | Lower uncertainty limit (m$^3$ s$^{-1}$ km$^{-2}$) | Upper uncertainty limit (m$^3$ s$^{-1}$ km$^{-2}$) | Modeled peak discharge (m$^3$ s$^{-1}$ km$^{-2}$) | Used for calibration (C) |
|---|---|---|---|---|---|---|---|---|---|
| 1 | Mamshit | 59 | H | 25/04/2018 | 1.5 | 1.3 | 1.6 | 0.4 | C |
| 2 | Zin Mashosh | 674 | H | 25/04/2018 | 0.0 | 0.0 | 0.0 | 0.2 | |
| 3 | Zin Avdat | 125 | H | 25/04/2018 | 0.0 | 0.0 | 0.0 | 0.0 | |
| 4 | Hatira crater | 48 | P | 25/04/2018 | 0.9 | 0.8 | 0.9 | 1.0 | C |
| 5 | Zafit 1 | 36 | P | 26/04/2018 | 1.3 | 1.1 | 1.5 | 1.1 | C |
| 6 | Zafit 2 | 41 | P | 26/04/2018 | 1.9 | 1.7 | 2.1 | 1.6 | C |
| 7 | Zafit 3 | 46 | P | 26/04/2018 | 1.7 | 1.5 | 1.8 | 1.4 | C |
| 8 | Tamar 1 | 4 | P | 26/04/2018 | 2.8 | 1.2 | 4.7 | 2.9 | C |
| 9 | Oretz | 5 | P | 26/04/2018 | 8.2 | 7.8 | 9.2 | 7.0 | C |
| 10 | Mazar | 3 | P | 26/04/2018 | 6.4 | 5.7 | 6.8 | 5.0 | C |
| 11 | Mamshit | 59 | H | 26/04/2018 | 1.3 | 1.1 | 1.4 | 0.5 | C |
| 12 | Zin Mashosh | 674 | H | 26/04/2018 | 0.1 | 0.1 | 0.1 | 0.0 | |
| 13 | Zin Avdat | 125 | H | 26/04/2018 | 0.0 | 0.0 | 0.0 | 0.0 | |
| 14 | Mamshit | 59 | H | 27/04/2018 | 1.7 | 1.5 | 1.9 | 0.3 | C |
| 15 | Zin Mashosh | 674 | H | 27/04/2018 | 0.8 | 0.7 | 0.8 | 1.2 | |
| 16 | Zin Mapal | 234 | H | 27/04/2018 | 0.0 | 0.0 | 0.0 | 0.0 | |
| 17 | Zin Avdat | 125 | H | 27/04/2018 | 0.0 | 0.0 | 0.0 | 0.0 | |
| 18 | Yamin | 31 | P | 27/04/2018 | 4.3 | 3.9 | 4.7 | 4.1 | C |
| 19 | Hatzera | 59 | P | 27/04/2018 | 1.6 | 1.4 | 1.8 | 0.7 | C |

Table 2: Zin sub-basin properties, peak discharge for return periods using GEV fit, and the estimated flash-flood return period category based on the modelled peak discharge of April 26[th]

| General properties | | Outlet coordinates (ITM coordinates) | | Main channel | | Storm peak discharge thresholds for return period of ($m^3 s^{-1} km^{-2}$) | | | Flash flood return period category |
|---|---|---|---|---|---|---|---|---|---|
| Sub-basin name | Area ($km^2$) | X | Y | Length (km) | Gradient ($m\ m^{-1}$) | 2 years | 10 years | 50 years | 26/04/2018 |
| Zafit 1 | 36 | 225546 | 543965 | 17 | 0.03 | 0.1 | 1.0 | 2.9 | Large |
| Zafit 2 | 41 | 225934 | 543026 | 18 | 0.03 | 0.1 | 0.9 | 2.7 | Large |
| Zafit 3 | 46 | 228782 | 542285 | 22 | 0.03 | 0.1 | 0.8 | 2.5 | Large |
| Tamar 1 | 4 | 225583 | 545041 | 4 | 0.09 | 0.5 | 3.6 | 10.7 | Moderate |
| Tamar 2 | 13 | 231184 | 543857 | 11 | 0.06 | 0.3 | 1.8 | 5.5 | Moderate |
| Peres 1 | 11 | 225004 | 549725 | 7 | 0.02 | 0.3 | 2.0 | 6.1 | Moderate |
| Peres 2 | 27 | 227856 | 545777 | 13 | 0.04 | 0.2 | 1.2 | 3.5 | Moderate |
| Peres 3 | 33 | 231121 | 543931 | 17 | 0.04 | 0.1 | 1.0 | 3.1 | Moderate |
| Oretz | 5 | 225442 | 543723 | 5 | 0.07 | 0.5 | 3.3 | 10.0 | Large |
| Mazar | 3 | 223644 | 542280 | 4 | 0.08 | 0.6 | 4.4 | 13.1 | Large |
| Tznim | 5 | 226533 | 537850 | 5 | 0.04 | 0.5 | 3.2 | 9.5 | Low |
| Hatzera crater | 48 | 220789 | 540072 | 8 | 0.06 | 0.1 | 0.8 | 2.4 | Moderate |
| Mitzlaot | 4 | 217589 | 534839 | 5 | 0.08 | 0.5 | 3.5 | 10.6 | Moderate |
| Akrabim | 6 | 216853 | 534077 | 6 | 0.08 | 0.4 | 3.0 | 8.9 | Large |
| Gov | 9 | 212688 | 532776 | 8 | 0.06 | 0.3 | 2.3 | 6.8 | Large |
| Koshesh | 11 | 211487 | 531617 | 9 | 0.02 | 0.3 | 2.0 | 6.1 | Low |
| Hatira | 270 | 210785 | 531523 | 33 | 0.02 | 0.0 | 0.3 | 0.8 | Large |
| Taban | 12 | 210090 | 530495 | 9 | 0.02 | 0.3 | 1.9 | 5.7 | Low |
| Saraf | 11 | 202042 | 524775 | 8 | 0.03 | 0.3 | 2.1 | 6.2 | Low |
| Hagor | 18 | 201793 | 524727 | 12 | 0.03 | 0.2 | 1.5 | 4.4 | Low |
| Teref | 35 | 197295 | 520076 | 14 | 0.03 | 0.1 | 1.0 | 2.9 | Moderate |
| Deres | 6 | 196977 | 519843 | 5 | 0.06 | 0.4 | 3.0 | 8.9 | Low |
| Hava 3 | 80 | 197104 | 520922 | 36 | 0.02 | 0.1 | 0.6 | 1.8 | Low |
| Hava 2 | 43 | 189723 | 515033 | 24 | 0.01 | 0.1 | 0.9 | 2.6 | Low |
| Hava 1 | 15 | 186140 | 508042 | 11 | 0.01 | 0.2 | 1.6 | 4.9 | No flow |
| Znim | 47 | 197152 | 522668 | 15 | 0.03 | 0.1 | 0.8 | 2.4 | Low |
| Zarhan | 35 | 194734 | 525092 | 14 | 0.02 | 0.1 | 1.0 | 2.9 | Low |
| Ofran | 9 | 193432 | 528405 | 6 | 0.06 | 0.3 | 2.3 | 7.0 | Low |
| Mador | 16 | 193533 | 528225 | 9 | 0.03 | 0.2 | 1.6 | 4.9 | Low |
| Talul | 15 | 191839 | 526447 | 7 | 0.05 | 0.2 | 1.6 | 4.9 | Low |
| Zakuf | 5 | 191299 | 526330 | 6 | 0.05 | 0.5 | 3.4 | 10.2 | Low |
| Zik | 23 | 189421 | 525880 | 10 | 0.04 | 0.2 | 1.3 | 3.8 | Low |
| Daroch | 6 | 186241 | 527759 | 7 | 0.05 | 0.4 | 2.9 | 8.7 | No flow |
| Akev | 55 | 181785 | 527643 | 27 | 0.02 | 0.1 | 0.7 | 2.2 | No flow |
| Divshon | 15 | 179996 | 527870 | 10 | 0.03 | 0.2 | 1.7 | 5.0 | No flow |
| Havarim | 9 | 178584 | 528331 | 6 | 0.02 | 0.3 | 2.2 | 6.7 | Low |
| Rahatz | 5 | 177091 | 523203 | 6 | 0.02 | 0.5 | 3.3 | 10.0 | No flow |
| Retamim | 5 | 176562 | 523176 | 4 | 0.02 | 0.5 | 3.2 | 9.7 | Low |
| Avdat | 89 | 176784 | 522139 | 15 | 0.02 | 0.1 | 0.6 | 1.7 | No flow |
| Nafha | 8 | 181436 | 516906 | 5 | 0.02 | 0.3 | 2.4 | 7.1 | No flow |
| Aricha | 15 | 177885 | 510922 | 5 | 0.02 | 0.2 | 1.7 | 5.0 | No flow |
| Zin 1 | 44 | 177479 | 509735 | 15 | 0.01 | 0.1 | 0.8 | 2.5 | No flow |
| Zin Arava road | 1238 | 228391 | 539352 | 119 | 0.01 | 0.0 | 0.1 | 0.3 | Moderate |
| Zin west Arava road | 1227 | 225739 | 537627 | 115 | 0.01 | 0.0 | 0.1 | 0.3 | Moderate |
| Mamshit | 59 | 204286 | 541038 | 20 | 0.02 | 0.1 | 0.7 | 2.1 | Moderate |
| Zin Mashosh | 674 | 198037 | 523528 | 73 | 0.01 | 0.0 | 0.2 | 0.5 | Low |
| Zin Mapal | 234 | 177336 | 525781 | 44 | 0.01 | 0.0 | 0.3 | 0.9 | Low |
| Zin Avdat | 125 | 178385 | 520923 | 37 | 0.01 | 0.1 | 0.4 | 1.3 | No flow |
| Avdat 1 | 42 | 172309 | 517579 | 8 | 0.02 | 0.1 | 0.9 | 2.6 | No flow |
| Matred | 22 | 172135 | 517700 | 10 | 0.01 | 0.2 | 1.3 | 3.9 | No flow |
| Hatira crater | 58 | 202336 | 540129 | 16 | 0.02 | 0.1 | 0.7 | 2.1 | Low |
| Kamus | 12 | 205837 | 538047 | 8 | 0.03 | 0.3 | 1.9 | 5.8 | Low |
| Yamin | 31 | 207730 | 540177 | 10 | 0.02 | 0.2 | 1.1 | 3.2 | Moderate |
| Maale | 9 | 207729 | 539978 | 7 | 0.02 | 0.3 | 2.3 | 6.8 | Large |
| Golhan | 22 | 207185 | 536321 | 11 | 0.03 | 0.2 | 1.3 | 3.9 | Low |
| Zin outlet | 1364 | 233608 | 545326 | 129 | 0.01 | 0.0 | 0.1 | 0.3 | Moderate |
| Hatzera | 56 | 223083 | 538185 | 12 | 0.05 | 0.1 | 0.7 | 2.2 | Moderate |

Table 3: Field observations and model simulation results for the 26[th] April flash flood Zafit sub-basin

| Observation | | GB-HYDRA model | |
|---|---|---|---|
| Process | Time (UTC+3) | Process | Time (UTC+3) |
| Rain initiation | 12:00 | Rain cell reaches observers | 12:00 |
| Hillslope runoff, no stream runoff | 12:05 | RCA and stream runoff were not identified | 12:05 |
| Rain stops | 12:10 | | |
| Rain re-starts | 12:50 | Convective rain cell approaches from the north | |
| Hillslope runoff | 12:57 | RCA identification | 13:00 |
| Stream flow | 13:00 | Stream flow at simulation point - (Figure 6) peak at 13:20 UTC+3 | 13:00 |

**Table A1: Properties used for the GB-HYDRA model**

| Landcover | % of Zin basin | CN value | Calibration range | Hillslope Manning roughness coefficient | Drainage rate (mm h$^{-1}$) | Notes |
|---|---|---|---|---|---|---|
| **Sands** | 5 | 75 | 55-90 | 0.01 | 40 | Hatzeva formation |
| **Colluvium/ Alluvium** | 4 | 75 | - | 0.02 | No | |
| **Quarry** | 3 | 79 | - | 0.02 | No | |
| **Sandstones** | 6 | 85 | 70-90 | 0.01 | No | Kurnub group |
| **Rocky desert** | 80 | 92 | 90-97 | 0.01 | No | Judea, Mt. Scopus, and Avdat groups |
| **Built area** | 2 | 98 | - | 0.013 | No | Roads, buildings |

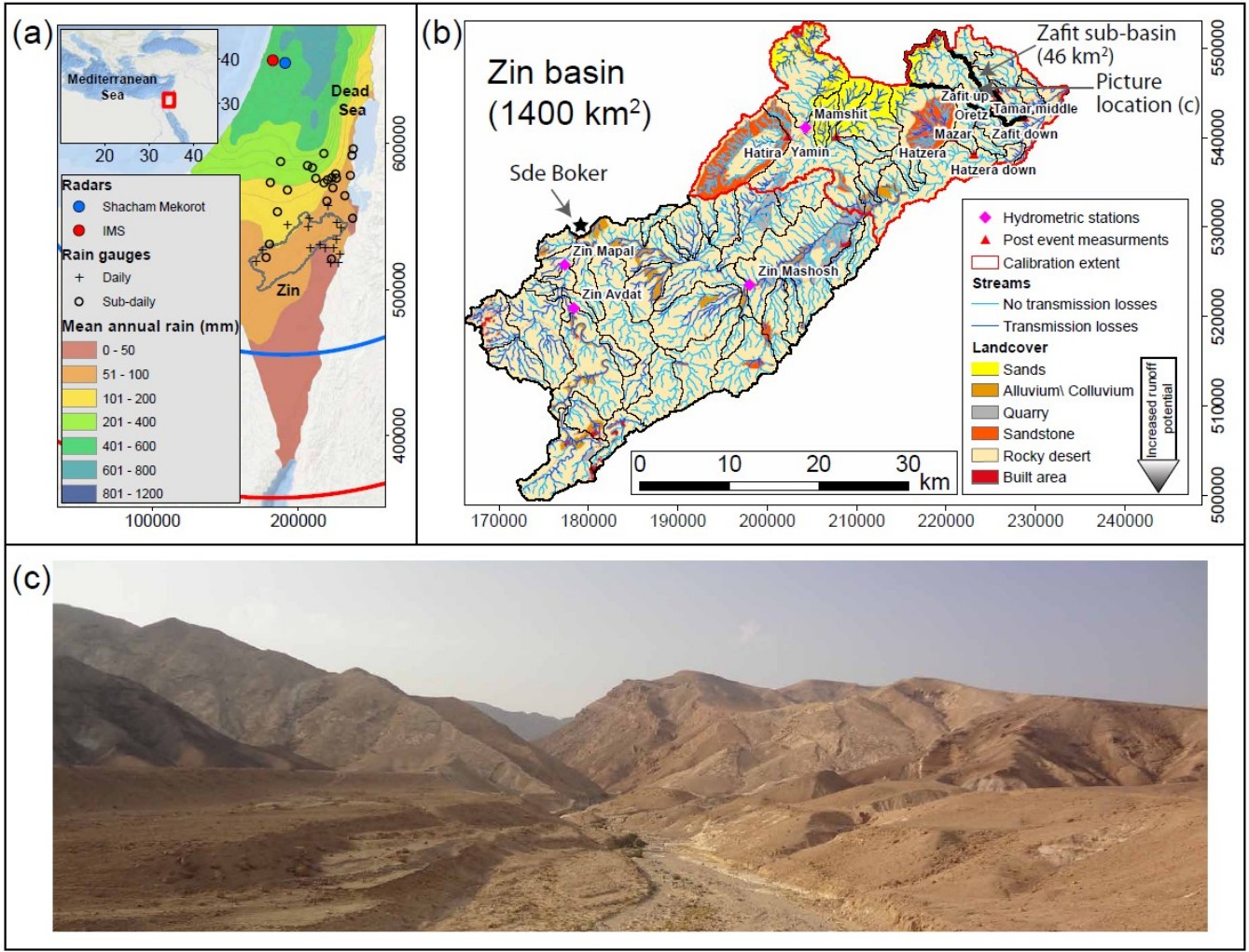


Figure 1: (a) Mean annual rain depth map of Israel (1960–1990, Israel Meteorological Service), daily and sub-daily rain gauges, and the Zin basin location. Extents and locations of the Shaham–Mekorot and the Israel Meteorological Service C-band radars are in blue and red, respectively. Inset: General location map. Map coordinates (in m) are of the Israeli Transverse Mercator grid, while for the inset, the geographic coordinate system is used. The World Ocean base map by Esri

is used at the background. (b) The Zin basin and its 57 sub-basins (thin brown lines). Zafit sub-basin is outlined in a bold black line. Hydrometric stations are shown as pink diamonds and locations of post-event estimations are marked by red triangles. (c) Photograph of Zafit basin taken toward the northwest, location is marked in panel b.


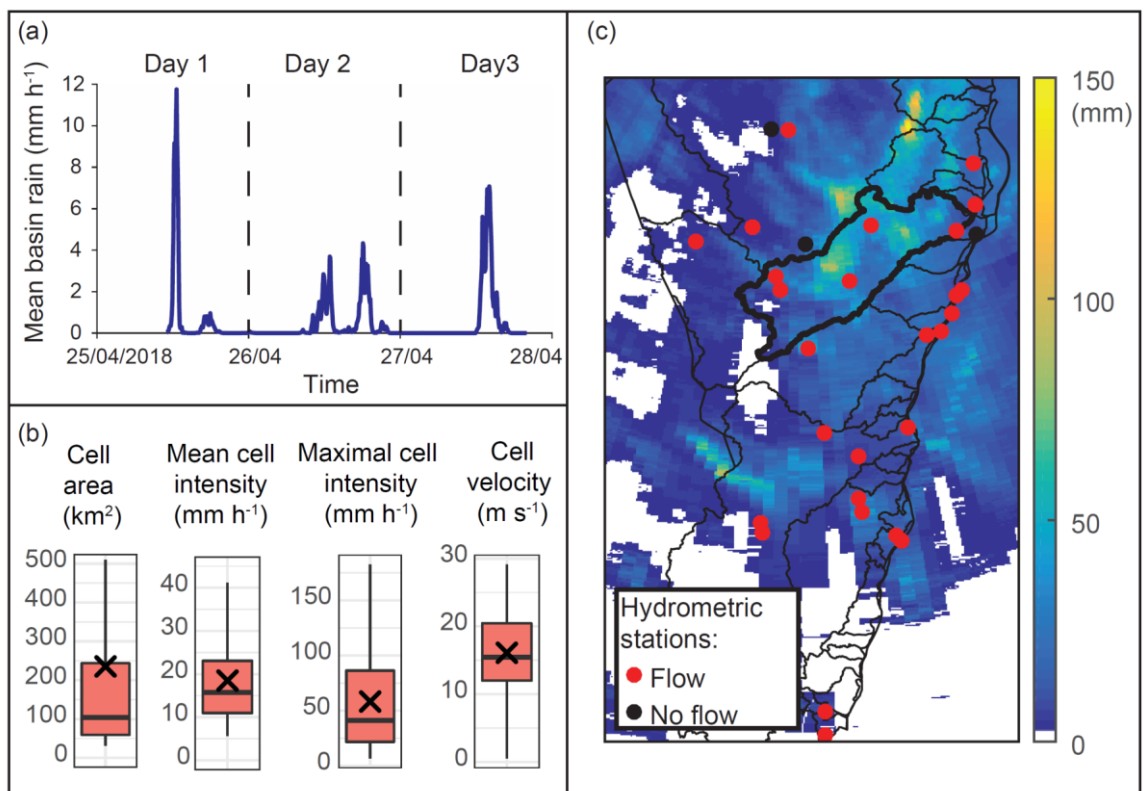

Figure 2: Properties of 25–27 Apr 2018 rainstorm. (a) 5-min rain intensity spatially averaged over the Zin basin. (b) Rain-cell area, mean and maximal rain-cell intensity, and rain-cell velocity calculated over the Zin basin. Crosses represent the mean, horizontal lines are the median, colored areas represent the narrowest 50% of the data, and the whisker limits represent the minimal and maximal values, except where 150% of the interquartile range is exceeded. (c) Total storm rain depth from the radar analysis, and location of hydrometric stations. Stations that exhibited at least one flash flood during the rainstorm are marked in red. The Zin basin is marked by a bold outline.


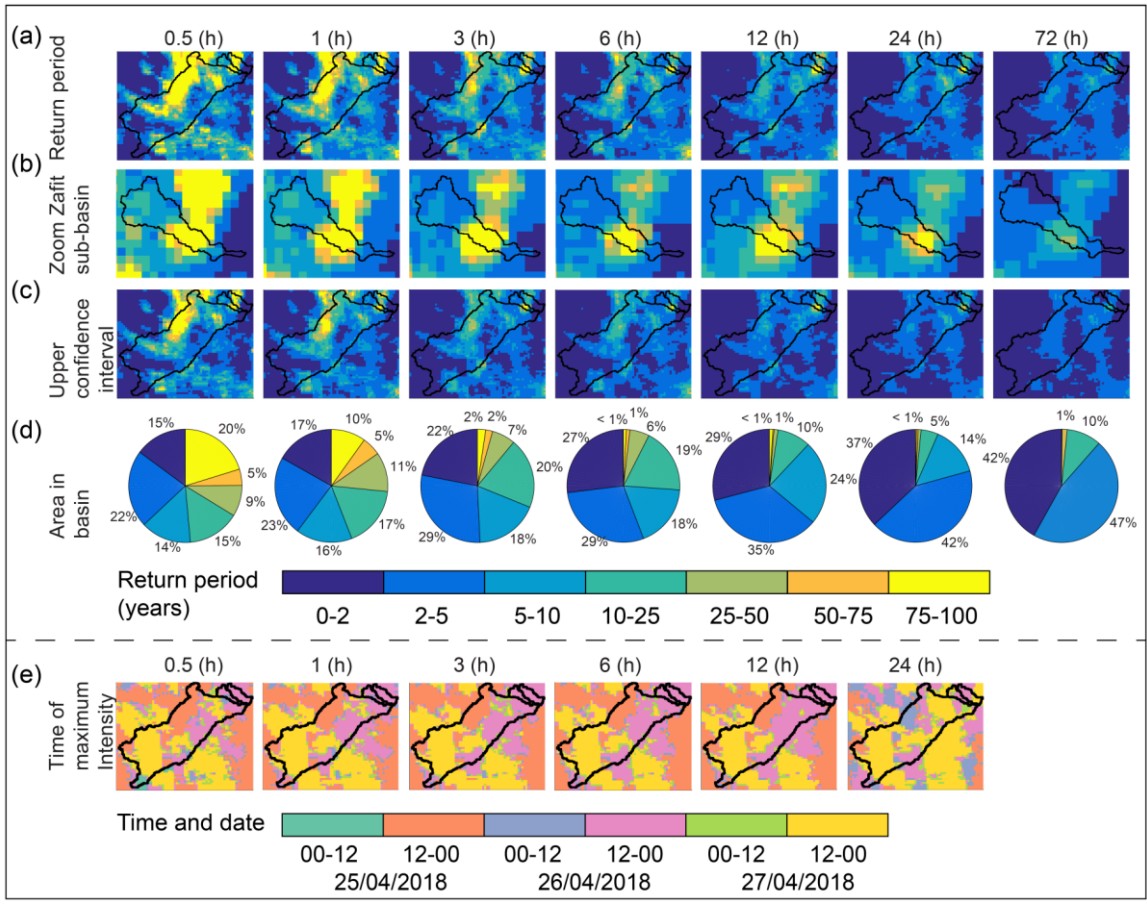

Figure 3: (a) Spatial return period maps for the studied storm. (b) Same as panel a, zoomed in to the Zafit sub-basin. (c) Upper confidence interval (95%) for the spatial return period maps. (d) Distribution of return periods in the Zin basin. (e) Timing of the maximal rainfall intensities, which was used to produce panels a–d. All properties are presented for durations of 0.5–72 h. Zin basin and Zafit sub-basin are outlined in black.

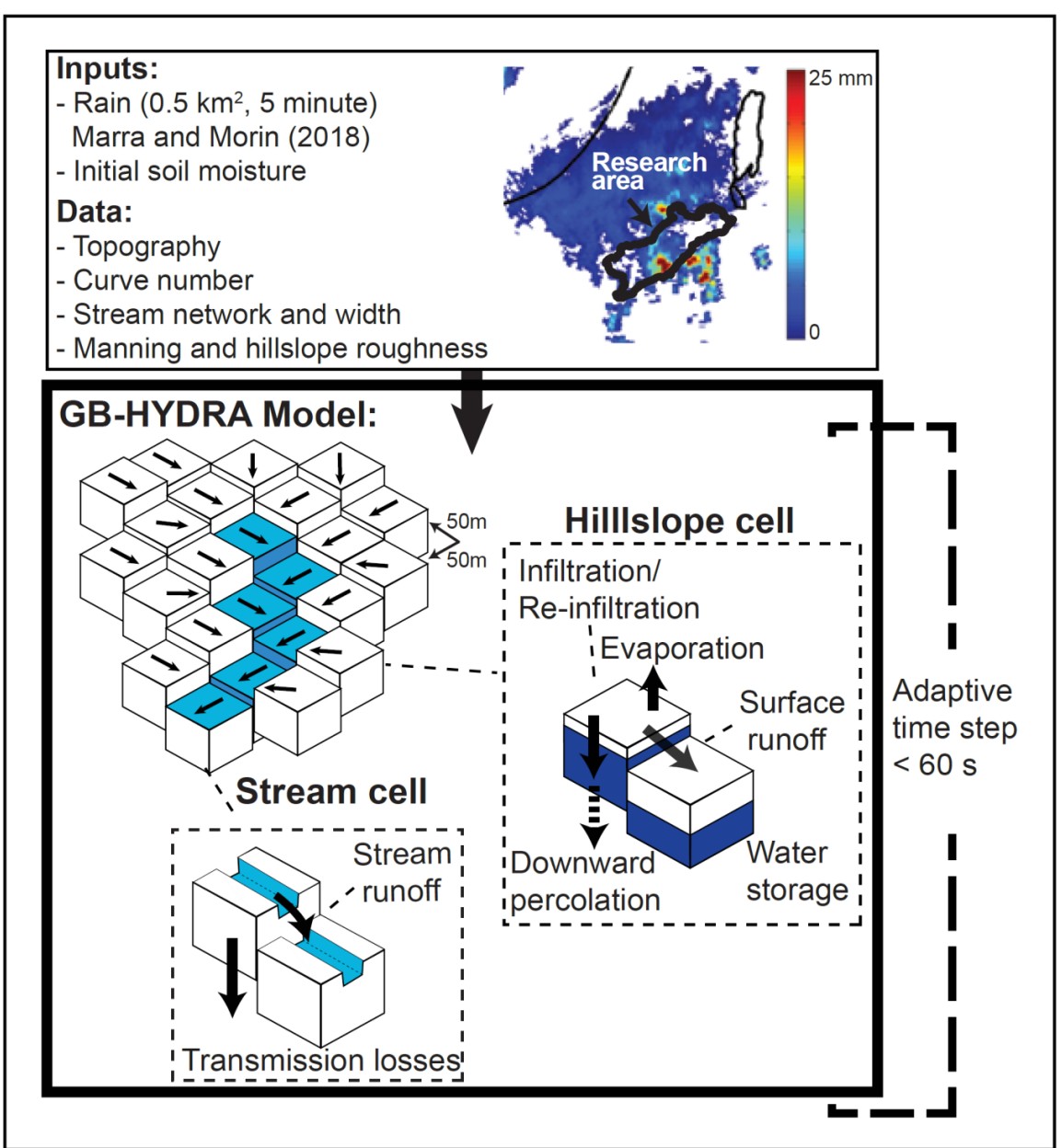

Figure 4: The modified GB-HYDRA model scheme, processes, and inputs. Further details are given in Appendix A and in
Rinat et al. (2018).

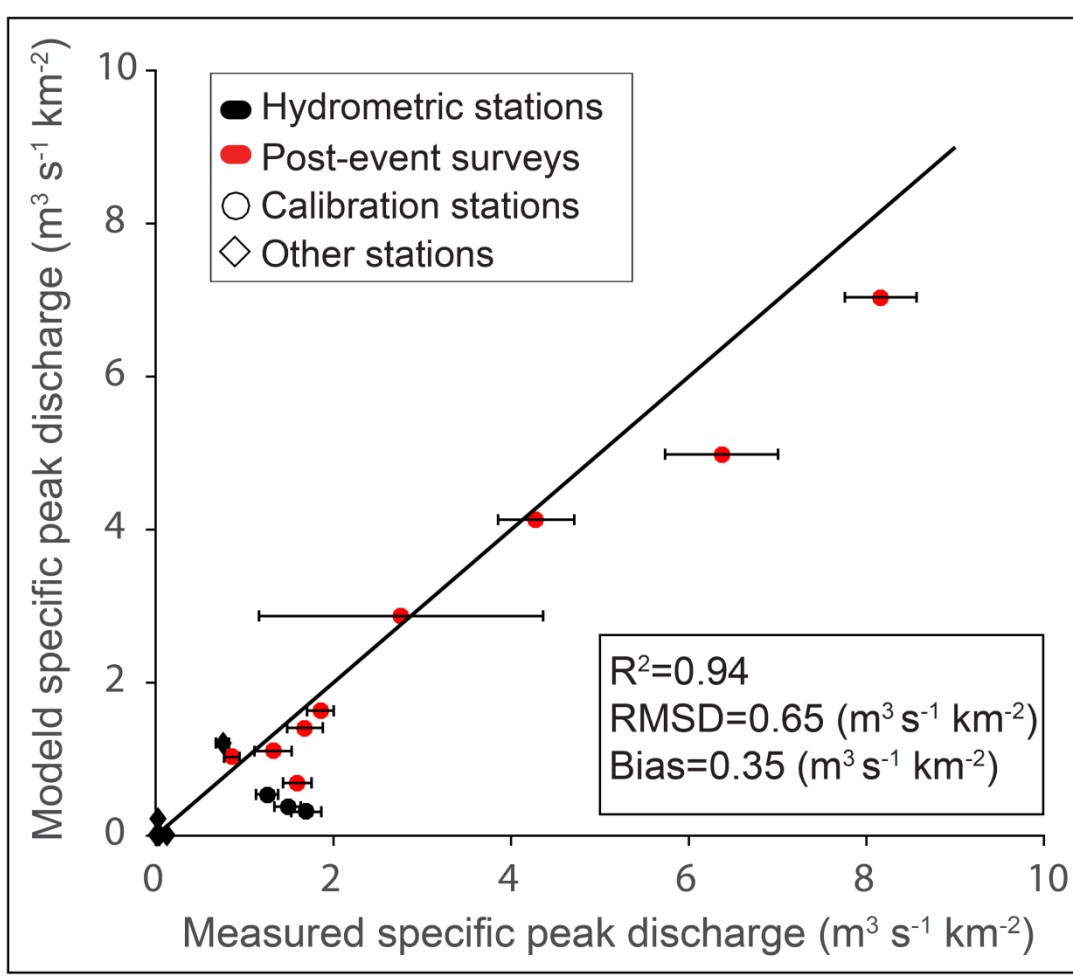


Figure 5: Measured vs. modeled specific peak discharge and error metrics for 12 peak discharges during the 3-day period of the storm. Horizontal error bars represent measured peak discharge uncertainty. Sub-basins considered in the calibration process are from the eastern side of the catchment (Fig. 1).


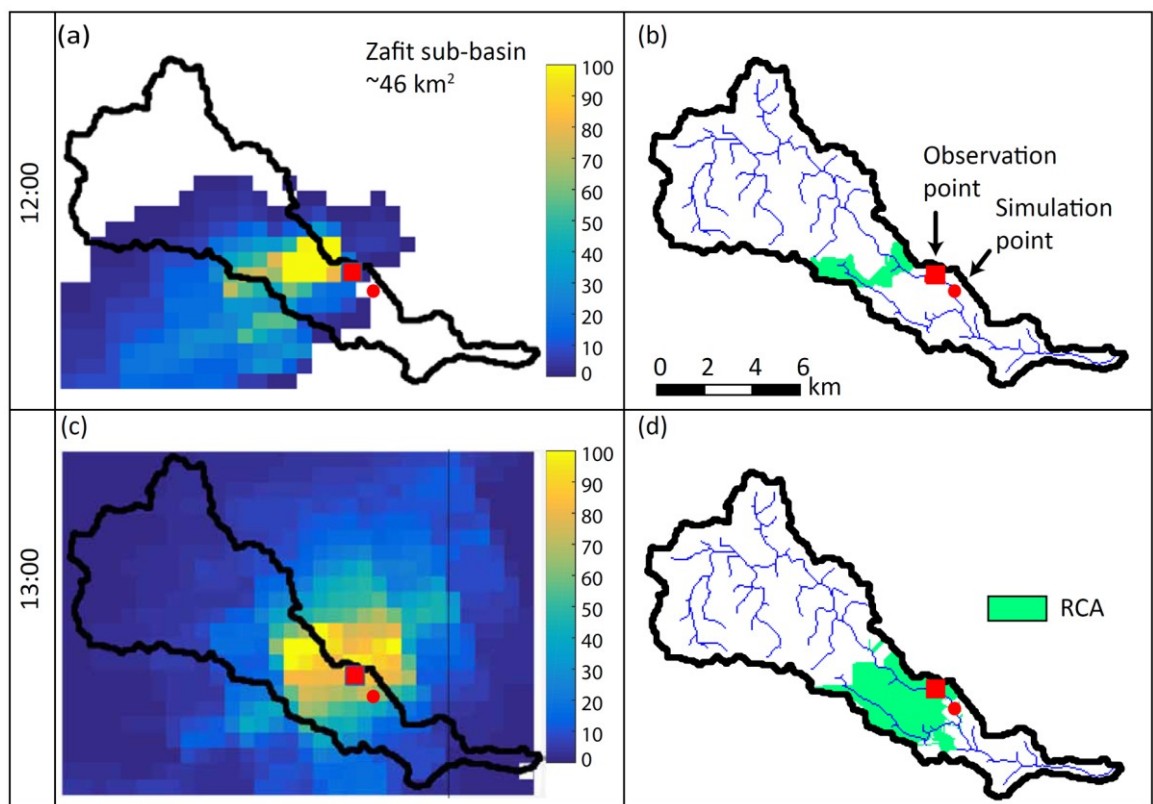

Figure 6: (a,c) 10-min rainfall maps and (b,d) RCA extent for the Zafit sub-basin at 12:00 and 13:00 (on April 26th). Red rectangles represent the observation point and red circle represents the simulation point for the spatial validation (Table 3), lag-time calculations, and Video 1.


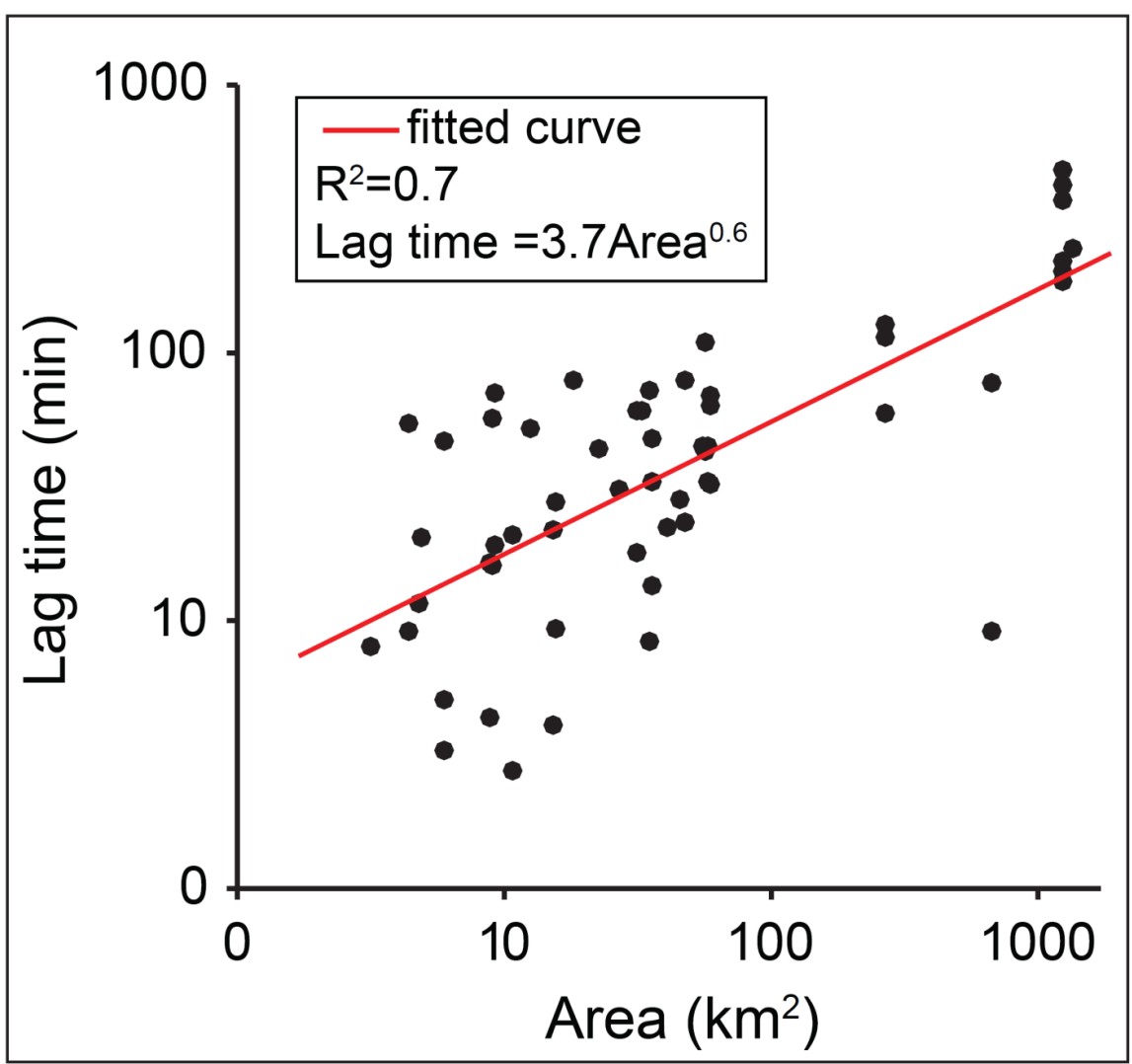


Figure 7: Modeled lag time versus basin area for all flash floods with peak discharge $>5 \text{ m}^3 \text{ s}^{-1}$.

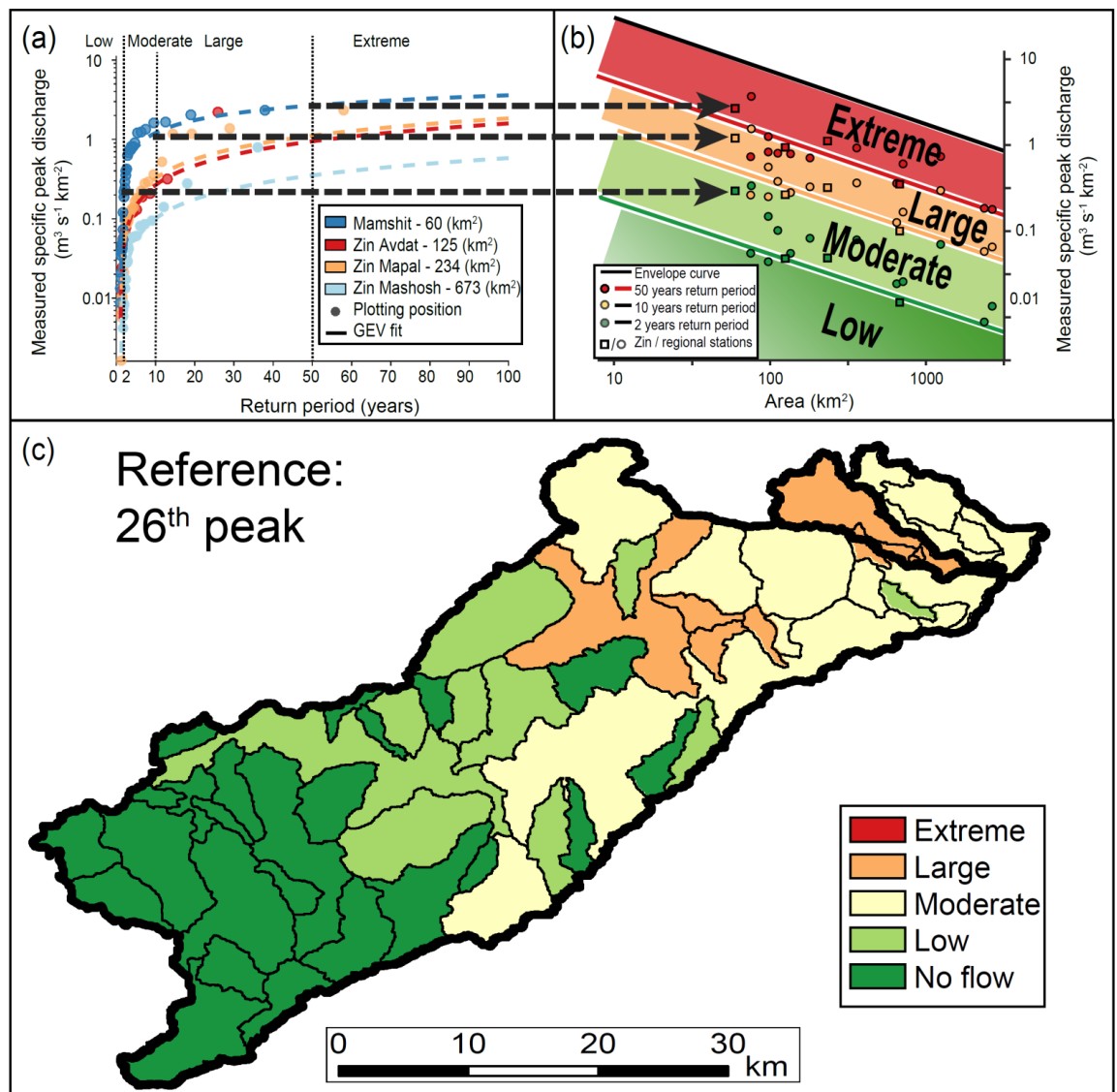

Figure 8: Determining flash-flood categories for each sub-basin. (a) Thresholds for each return period category were defined by applying GEV analysis on annual series of measured specific peak-discharge values from 18 hydrological stations (only 4 are shown) and projected on an area-specific peak-discharge domain (panel b). (b) The projected points from all stations and the local envelope curve (Tarolli et al., 2012) were used to define return period categories for each area. (c) April 26th flash-flood return period categories were calculated for all sub-basins (Zafit sub-basin is marked by bold outline).


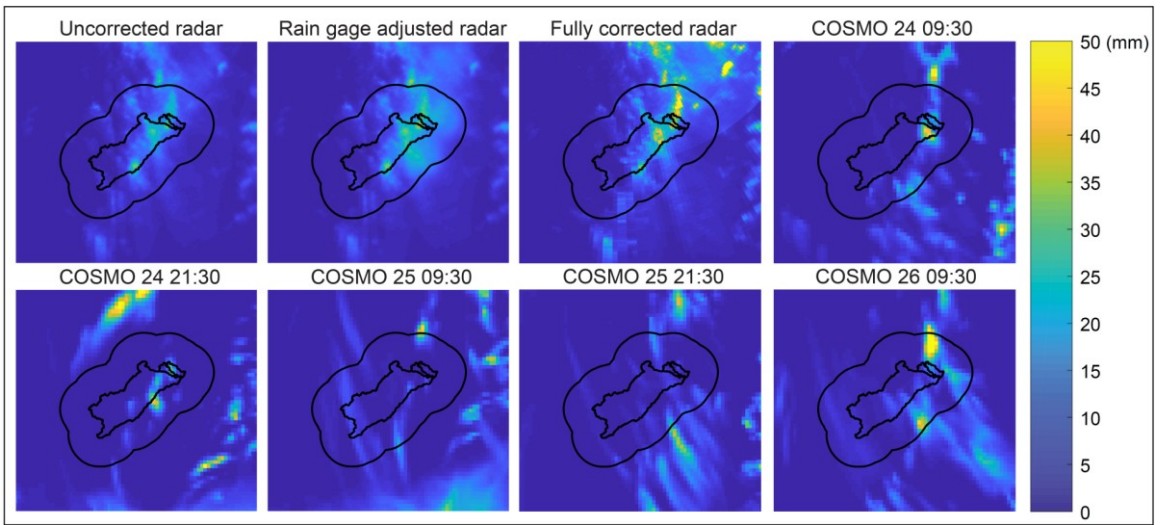

Figure 9: Total rain depth for Apr 26, 06:00 to 24:00, using different degrees of corrections to the radar data (three upper-left panels), and various COSMO runs (COSMO titles refers to the timing of forecast availability). The outlines of the Zin and Zafit basins are depicted, as well as a 20 km buffer zone around the Zin basin.


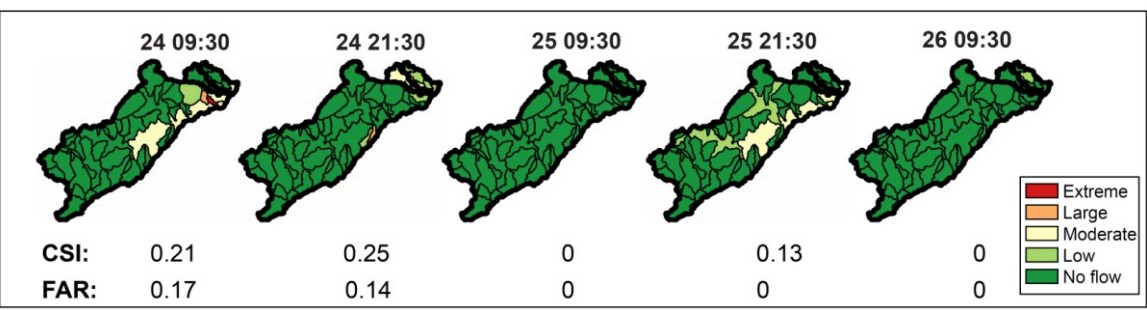

Figure 10: Flash-flood return period categories, CSI, and FAR, calculated using different COSMO forecast runs from the days preceding the April 26[th] flash flood. Headers state the time of forecast availability.


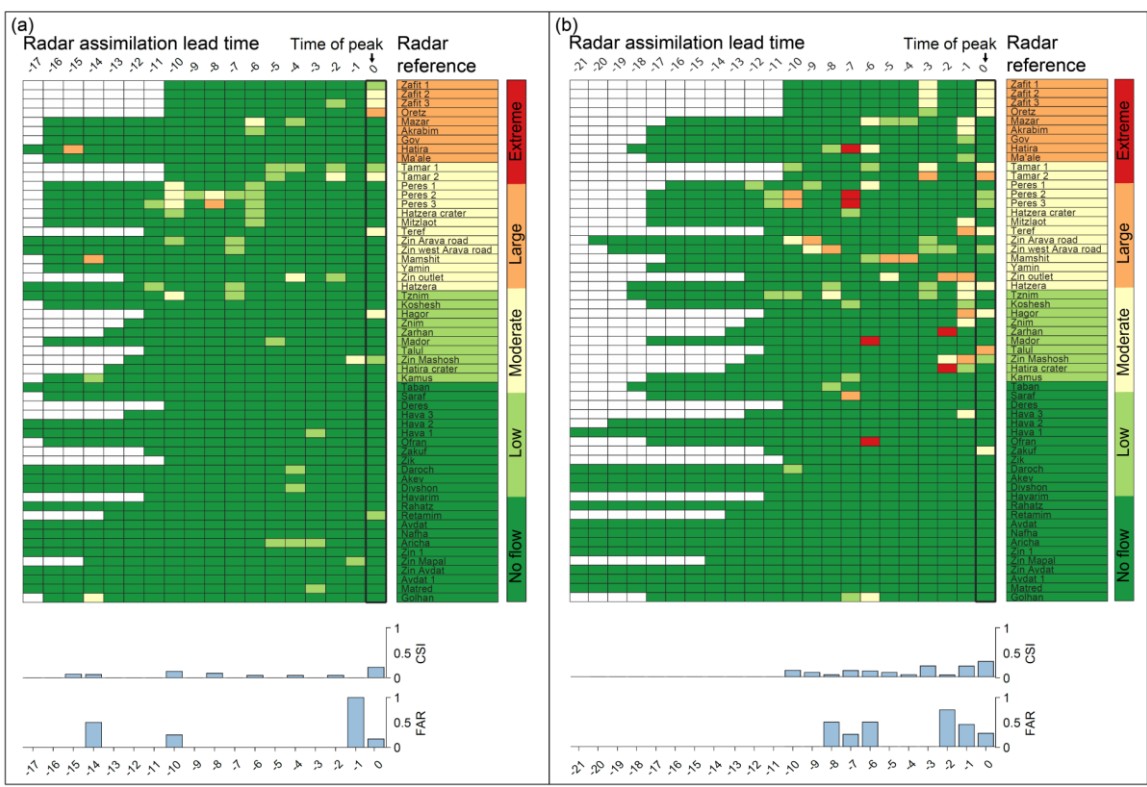


Figure 11: Flash-flood return period categories calculated for each sub-basin using the Apr 25, 19:30 (a) and Apr 26, 09:30 (b) COSMO runs with different radar assimilation lead times. Numbers represent the time in hours prior to the April 26[th] peak discharge at each sub-basin, i.e., the lead time (where 0 is the hour of the peak, columns are not synchronal). Radar reference, CSI, and FAR metrics are presented.


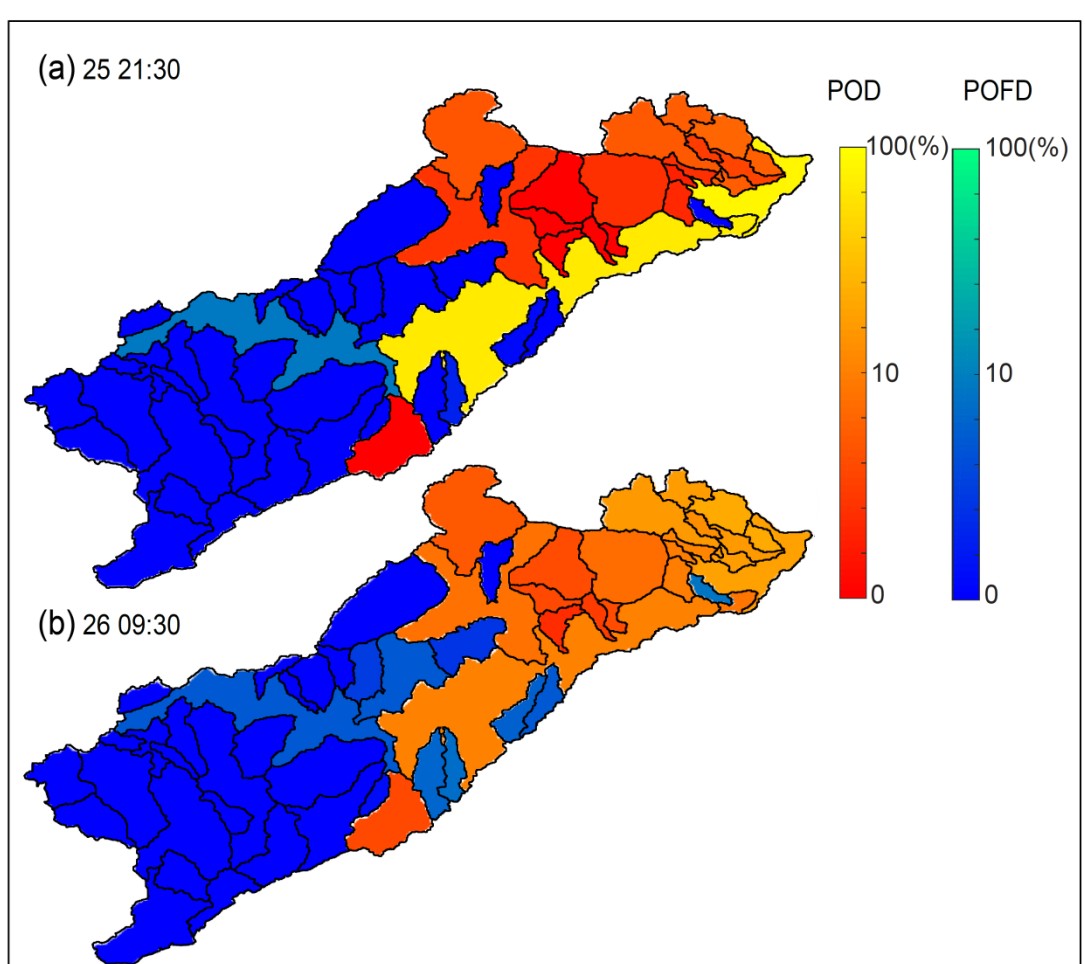

Figure 12: POD and POFD calculated for each sub-basin between the reference run and the peak discharge on April 26<sup>th</sup> for the 1681 rainfall fields resulting from the spatial shifting of the forecast COSMO runs of (a) Apr 25, 21:30 and (b) Apr 26, 09:30. POD was calculated for each sub-basin with reference category of moderate or above and PODF was calculated for each sub-basin with reference categories of low or no flow (Fig. 8).