# Peer review of "Hydrometeorological analysis and forecasting of a 3-day flash-flood-triggering desert rainstorm"

_Natural Hazards and Earth System Sciences, 2020_

## Referee Comment (RC1) · Anonymous Referee #1 · 31 Jul 2020

General comments The paper presents a detailed hydrometerological analysis of a flash flood that occurred in an arid area of Israel. The study also assesses the value of meteorological forecast for the early warning of such a devastating event. The paper is very well written and very clear. The study relies on an exceptionally rich data set including radar rainfall, post event survey estimation of peak discharge and a well-calibrated distributed hydrological model. The paper also capitalizes on previous work that characterized Intensity Duration Frequency of rainfall using a long archive of 5 min well calibrated radar rainfall fields. All the available information allows describing the spatio-temporal variability of the event and of the hydrological response. The paper interestingly shows that the return period of the flash flood can be much smaller from

that of the rainfall, due to a low connectivity of generated runoff with the river course. The paper lacks a discussion section that could be useful to better highlight the interest of the present study as compared to the existing literature. I also have some minor comments that are detailed below. After addressing them and adding a discussion section the paper will be acceptable for publication in Natural Hazard and Earth System Science.

Specific comments 1/ Line 122: the paper refers to Table 1, but this table does not seem to be related to the sentence 2/ Line 143-145: The authors could mention the general methodology for post event survey proposed by Gaume and Borga (2008) 3/ Lines 190-194: the use of the bootstrap method is not clear: on which variable is the bootstrapping performed? 4/ Line 224: the authors mention that the video provided by the witnesses gives information on the spatial and temporal variability. I would say that it only gives information on the temporal variability as the witnesses are located at only one point. 5/ Line 262 The model resolution is . . . 6/ Lines 273-274 and lines 320-321. The authors should refer to the work of Vincendon et al. (2011) that already proposed a method to move the location of intense cells from deterministic meteorological forecast. The method of Vincendon et al. (2011) is very close to the method proposed in the present paper. 7/ Table 1: The content of this table is not clear: does it include the estimated peak discharge from the post event survey? References: Gaume, E., Borga, M., 2008. Post-flood field investigations in upland catchments after major flash floods: proposal of a methodology and illustrations. Journal of Flood Risk Management, 1(4): 175-189. DOI:10.1111/j.1753-318X.2008.00023.x Vincendon, B., Ducrocq, V., Nuissier, O., Vie, B., 2011. Perturbation of convection-permitting NWP forecasts for flash-flood ensemble forecasting. Natural Hazards and Earth System Sciences, 11(5): 1529-1544.

---

## Referee Comment (RC2) · Lorenzo Marchi (Referee) · 24 Sep 2020

This paper provides a valuable contribution to the assessment of flash-flood response and the performance of precipitation forecasts in arid watersheds. I am reporting below some comments for paper revision.

The post-flood reconstruction of peak discharge is affected by several sources of errors, including measurement errors and uncertainties in the estimation of the roughness coefficient. The authors could consider assessing and presenting (Table 2 and Fig. 5) the uncertainties bounds of post-flood peak discharge estimates. Indirect estimates of flash flood peak discharge, especially if validated by a rainfall-runoff model,

like in this work, are of utmost importance for getting a better knowledge of these hazardous phenomena, also for comparison with other datasets. Reporting the uncertainties bounds increases the value of such flood peak data.

Section 4.1.1, which reports field observations by two scientists who witnessed the flood at the Zafit sub-basin, could be extended, for instance by describing the main geomorphic effects of the flood. The title of this section could be modified for emphasizing that it contains direct observations of the flash flood.

Lines 54-56. Not only in arid regions: also under humid climates, the strong spatial gradients of rainfall fields make the rain gauge network inadequate to represent flash flood triggering rainfall.

Line 128. The area of the Zafit sub-basin (46 km$^2$ - line 91) could be recalled here.

Lines 136-137. The absence of rain gauges within the basin (cf. Fig. 1 and lines 169-170) should be clearly stated.

Lines 140-141. "however, only one of these monitors the area influenced by the storm's core": which one (cf. table 2)?

Lines 374-376 and Table A1. Quite low values of Manning roughness coefficients for hillslopes. The works by Downer and Ogden (2002), Engman (1986), and Sadeh et al. (2018), which apparently support these values, are not reported in the references list.

Line 335. "Rain gauges in desert areas fail to represent the spatial heterogeneity of convective rainfall". In general, this statement sounds rather convincing. In the case of the April 2018 flash flood in the Zin basin, however, the only rain gauge available was located outside the basin, so that no conclusion on the suitability of rain gauge data can be drawn.

Line 337. "whereas"?

Lines 651-652. The final paper, instead of the discussion version, should be reported.

Table 1 lists properties and flood response of 57 sub-basins that in which the flood of April 2018 was analyzed using the GB-HYDRO model. It is not clear why this table is cited in section 2.1, which describes the settings of the study region with a focus on past flood events.

Table 2: I suggest reporting the drainage basin area of the sub-basins.

---

## Author Comment (AC1) · 18 Oct 2020

We thank Reviewer #1 for the useful comments that helped us improve the manuscript. We addressed all comments in a revised manuscript we have prepared. The reviewer's comments are reported followed by our answers.

- The paper lacks a discussion section that could be useful to better highlight the interest of the present study as compared to the existing literature. Authors' answer: Discussion section was added as suggested. It includes the lines of discussion that were previously distributed in the individual sections as well as some additional lines.

Specific comments: - 1/ Line 122: the paper refers to Table 1, but this table does not seem to be related to the sentence Authors' answer: Thank you the reference was removed, consequently, the places of Table 1 and Table 2 were swapped

- 2/ Line 143-145: The authors could mention the general methodology for post event survey proposed by Gaume and Borga (2008). Authors' answer: Thank you, the reference was added

- 3/ Lines 190-194: the use of the bootstrap method is not clear: on which variable is the bootstrapping performed? Authors' answer: The sentence has been updated to: "Uncertainty related to the available data record was quantified via bootstrap with replacement (250 repetitions) among the years in the record"

- 4/ Line 224: the authors mention that the video provided by the witnesses gives information on the spatial and temporal variability. I would say that it only gives information on the temporal variability as the witnesses are located at only one point. Authors' answer: Thank you, it was corrected

- 5/Line 262 The model resolution is : Authors' answer: Sentence was changed from: "The model resolution is of $\sim$2.5 km2" to: The model spatial resolution is of $\sim$2.5 km.

- 6/ Lines 273-274 and lines 320-321. The authors should refer to the work of Vincendon et al. (2011) that already proposed a method to move the location of intense cells from deterministic meteorological forecast. The method of Vincendon et al. (2011) is very close to the method proposed in the present paper. Authors' answer: Thanks, it was added to the new discussion: "Vincendon et al., (2011) developed a simplified method to produce rainfall ensemble from single-value meteorological forecasts and showed that by using it as an input to a hydrological model the gained flood-forecasting ensemble performs better than the deterministic result."

- 7/ Table 1: The content of this table is not clear: does it include the estimated peak discharge from the post event survey? Authors' answer: No, peak discharge data from

post event survey and from hydrometric stations are in Table 2 (now Table 1). Table 1 (now Table 2) includes general sub-basins properties, discharge thresholds that are used to determine flash flood return period category, and the determined flood return period category for April 26th; text was added for clarification.

---

## Author Comment (AC2) · 18 Oct 2020

Lorenzo Marchi, referee #2

We thank Lorenzo Marchi for the useful comments that helped us improve the manuscript. We addressed all comments in a revised manuscript we have prepared. The reviewer's comments are reported followed by our answers.

- The post-flood reconstruction of peak discharge is affected by several sources of errors, including measurement errors and uncertainties in the estimation of the roughness coefficient. The authors could consider assessing and presenting (Table 2 and

[Figure]

Fig. 5) the uncertainties bounds of post-flood peak discharge estimates. Indirect estimates of flash flood peak discharge, especially if validated by a rainfall-runoff model, like in this work, are of utmost importance for getting a better knowledge of these hazardous phenomena, also for comparison with other datasets. Reporting the uncertainties bounds increases the value of such flood peak data. Authors' answer: Uncertainty limits were added to Table 1 (previously Table 2) and Fig. 5 as suggested. Table 1 now contains 2 new columns: "Lower uncertainty limit" and "Upper uncertainty limit". Three types of data sets were used for all measured peak discharges: Type 1: six post event peak discharge estimates were derived by our team. For calculating the flood peak discharge, we surveyed post event high water marks such as drift wood and banks erosion lines. Topographic surveys of each of these reaches included the geometry of several cross-sections, longitudinal channel profiles and precise elevation measurements of the water marks. Discharge estimates were calculated by hydraulic modeling of the surveyed study reaches using HEC-RAS software. The uncertainties range of these estimations is now presented in the table. Type 2: three post event peak discharge estimates were derived by the Israel Hydrological Service and by the Soil and Erosion Research Station – both teams assess the uncertainty to be ~10%. Type 3: ten peak discharge estimates were obtained from hydrographs produced by the Israel Hydrological Service from hydrometric station data. Uncertainty is also estimated here as 10%.

- Section 4.1.1, which reports field observations by two scientists who witnessed the flood at the Zafit sub-basin, could be extended, for instance by describing the main geomorphic effects of the flood. The title of this section could be modified for emphasizing that it contains direct observations of the flash flood. Authors' answer: Unfortunately, we do not have further insights from the observations, such as the geomorphic effects. The title has been changed as suggested to: "Using direct observations for spatial model validation and flash-flood initiation"

- Lines 54-56. Not only in arid regions: also under humid climates, the strong spatial

gradients of rainfall fields make the rain gauge network inadequate to represent flash flood triggering rainfall. Authors' answer: Corrected as suggested.

- Line 128. The area of the Zafit sub-basin (46 km2 - line 91) could be recalled here. Authors' answer: Thank you, the area was added

- Lines 136-137. The absence of rain gauges within the basin (cf. Fig. 1 and lines 169-170) should be clearly stated. Authors' answer: Thank you. The locations of all rain gauges were added to Fig1a as suggested, and a sentence was added for clarification: "Two rain gauges with temporal resolution of 10-min and eight rain gauges that provide only daily data monitor the basin"

- Lines 140-141. "however, only one of these monitors the area influenced by the storm's core": which one (cf. table 2)? Authors' answer: Corrected, the sentence has been changed to "however, only the Mamshit hydrometric station is situated at the area influenced by the storm's core (Fig 1b, Table 1)"

- Lines 374-376 and Table A1. Quite low values of Manning roughness coefficients for hillslopes. The works by Downer and Ogden (2002), Engman (1986), and Sadeh et al. (2018), which apparently support these values, are not reported in the references list. Authors' answer: Thanks, the reference list was corrected. Indeed, these numbers are quite low, but they are supported by several works including the ones listed above and Shmilovitz et al., 2020 (added to the reference list). Furthermore, the low hillslope roughness coefficient contributes to the fast runoff generation that is typical to arid areas and reported by eyewitnesses in this work.

- Line 335. "Rain gauges in desert areas fail to represent the spatial heterogeneity of convective rainfall". In general, this statement sounds rather convincing. In the case of the April 2018 flash flood in the Zin basin, however, the only rain gauge available was located outside the basin, so that no conclusion on the suitability of rain gauge data can be drawn. Authors' answer: Agree, and removed.

- Line 337. "whereas"? Authors' answer: Thank you, corrected

- Lines 651-652. The final paper, instead of the discussion version, should be reported. Authors' answer: Thank you for the hint. The reference has been updated.

- Table 1 lists properties and flood response of 57 sub-basins that in which the flood of April 2018 was analyzed using the GB-HYDRO model. It is not clear why this table is cited in section 2.1, which describes the settings of the study region with a focus on past flood events. Authors' answer: Agreed. The cross reference was removed. Consequently, he places of Table 1 and Table 2 were swapped.

- Table 2: I suggest reporting the drainage basin area of the sub-basins Authors' answer: Thank you, added
* * *
**(a)**

**Mediterranean Sea**

**Dead Sea**

**Radars**
- ● Shacham Mekorot
- ● IMS

**Rain gauges**
- + Daily
- ○ Sub-daily

**Mean annual rain (mm)**
- 0 - 50
- 51 - 100
- 101 - 200
- 201 - 400
- 401 - 600
- 601 - 800
- 801 - 1200

Zin

**(b)**

Zin basin (1400 km²)

Zafit sub-basin (46 km²)

Picture location (c)

Mamshit

Zafit up, Tamar middle
Oretz
Mazar, Zafit down
Hatzera
Hatzera down

Hatira, Yamin

Sde Boker

Zin Mapal

Zin Mashosh

Zin Avdat

**Hydrometric stations** ◆
**Post event measurments** ▲
□ Calibration extent

**Streams**
— No transmission losses
— Transmission losses

**Landcover**
- Sands
- Alluvium\ Colluvium
- Quarry
- Sandstone
- Rocky desert
- Built area

Increased runoff potential

0  10  20  30
km

**(c)**

**Fig. 1.** The corrected Fig 1 in the manuscript

[Figure]

**Fig. 2.** The corrected Fig 5 in the manuscript